# Single-cell transcriptomics identifies divergent developmental lineage trajectories during human pituitary development

Shu Zhang [1,2,6], Yueli Cui[1,2,6], Xinyi Ma [1,3,6], Jun Yong[1,3,4], Liying Yan [1,3], Ming Yang[1,3,4,5], Jie Ren[1,2], Fuchou Tang [1,2,5], Lu Wen [1,2✉] & Jie Qiao [1,2,3,4✉]

The anterior pituitary gland plays a central role in regulating various physiological processes, including body growth, reproduction, metabolism and stress response. Here, we perform single-cell RNA-sequencing (scRNA-seq) of 4113 individual cells from human fetal pituitaries. We characterize divergent developmental trajectories with distinct transitional intermediate states in five hormone-producing cell lineages. Corticotropes exhibit an early intermediate state prior to full differentiation. Three cell types of the PIT-1 lineage (somatotropes, lactotropes and thyrotropes) segregate from a common progenitor coexpressing lineage-specific transcription factors of different sublineages. Gonadotropes experience two multistep developmental trajectories. Furthermore, we identify a fetal gonadotrope cell subtype expressing the primate-specific hormone chorionic gonadotropin. We also characterize the cellular heterogeneity of pituitary stem cells and identify a hybrid epithelial/mesenchymal state and an early-to-late state transition. Here, our results provide insights into the transcriptional landscape of human pituitary development, defining distinct cell substates and subtypes and illustrating transcription factor dynamics during cell fate commitment.

[1] Department of Obstetrics and Gynecology, Beijing Advanced Innovation Center for Genomics, School of Life Sciences, Third Hospital, Peking University, Beijing 100871, China. [2] Biomedical Pioneering Innovation Center, School of Life Sciences, Ministry of Education Key Laboratory of Cell Proliferation and Differentiation, Beijing 100871, China. [3] Key Laboratory of Assisted Reproduction, Ministry of Education, Beijing 100191, China. [4] Beijing Key Laboratory of Reproductive Endocrinology and Assisted Reproductive Technology, Beijing 100191, China. [5] Peking-Tsinghua Center for Life Sciences, Academy for Advanced Interdisciplinary Studies, Peking University, Beijing 100871, China. [6] These authors contributed equally; Shu Zhang, Yueli Cui, Xinyi Ma. ✉email: wenlu@pku.edu.cn; jie.qiao@263.net

The pituitary is the central gland of the endocrine system for regulating multiple physiological processes including the stress response, body growth, reproduction and metabolism. Much of the control comes from five cell types of the anterior pituitary gland including corticotropes that secrete adrenocorticotrophic hormone (ACTH), somatotropes that produce growth hormone (GH), lactotropes that release prolactin (PRL), thyrotropes that produce thyroid-stimulating hormone (TSH) and gonadotropes that produce luteinizing hormone (LH) and follicle-stimulating hormone (FSH). The development of the anterior pituitary provides an excellent model system to elucidate mechanism of organogenesis[1]. The five hormone producing cell types develop in a stereotypical order from Rathke's pouch, an epithelial invagination of the oral ectoderm. Previous studies have identified various signaling pathways and transcription factors (TFs) participating in pituitary development. These include the SHH, BMP and FGF pathways for initiation of Rathke's pouch; PITX1, LHX3, HESX1 and PROP1 for early phase patterning; POU1F1 (also known as PIT-1) for differentiation of somatotropes, lactotropes and thyrotropes; TBX19 (also known as TPIT) for differentiation of corticotropes; and NR5A1 (also known as SF-1) for differentiation of gonadotropes[1–4]. More recently, studies have also raised interest in SOX2-positive pituitary stem cells by showing their capability of self-renewal and differentiation into all five types of endocrine cells[5,6].

However, our understanding of pituitary development, particularly human pituitary development, is not well defined. Genomic studies have been hampered by intermingling of different cell types in this relatively small organ[7,8]. Recent rapid progression in single-cell RNA sequencing (scRNA-seq) technologies provides an opportunity to comprehensively understand the regulatory network and cellular heterogeneity of pituitary development. Several recent studies have reported scRNA-seq in the adult mouse and rat pituitary[9–11]. Here, we apply scRNA-seq to human fetal pituitaries for mapping the transcriptional landscape of human pituitary development at single-cell resolution. Our results provide insights into transcriptional dynamics of progressive lineage specification of human pituitary endocrine cells, and elucidate characteristics of the pituitary stem cells, progenitor and precursor cells, and different endocrine cell types and subtypes.

## Result

**ScRNA-seq analysis of human pituitary development.** We obtained pituitaries from 21 human fetuses from 7 to 25 weeks postfertilization (11 females and 10 males) and performed a modified STRT-seq method on a total of 5181 cells, with 4113 high-quality cells being retained after filtration (Fig. 1a and Supplementary Fig. 1a). An average of 4506 genes and 86,497 transcripts (counted as unique molecular identifiers, UMIs) were detected in each cell (Supplementary Fig. 1c). The samples were detected with similar gene numbers and GAPDH expression across batches (Supplementary Fig. 1b, c). The morphology of the pituitary was verified (Supplementary Fig. 1g).

We used Seurat to identify cell clusters and Uniform Manifold Approximation and Projection (UMAP) for visualization (Fig. 1b and Supplementary Fig. 1d)[12]. A total of 14 cell clusters identified with known marker genes (Fig. 1b, Supplementary Fig. 1d). We identified nine clusters of anterior pituitary endocrine cells including the stem cells (Stem), cycling cells (CC), corticotropes, progenitors of the PIT-1 lineage (Pro.PIT1), somatotropes, lactotropes, thyrotropes, precursors of gonadotropes (Pre. Gonado) and gonadotropes, comprising 2,388 cells (Fig. 1b and Supplementary Fig. 1a). PITX1 and PITX2 were expressed in all nine clusters, and SOX2 and PROP1 were specifically expressed in

the stem cells (Fig. 1c and Supplementary Fig. 1f). Lineage-specific TFs, such as POU1F1, TBX19 and NR5A1, and hormone genes, including POMC, GH1, GH2, TSHB, PRL, FSHB and LHB, were expressed in special clusters of the endocrine cells. Mesenchymal cells, as marked by PDGFRA[13], were the second most abundant cell type after endocrine cells, comprising 1,005 cells. A cluster of 90 cells was identified as posterior pituitary pituicyte cells (P) as marked by OTX2, LHX2, RAX and COL25A1[11,14] (Supplementary Fig. 1f). Other clusters included endothelial cells (PECAM1+), immune cells (IMM, PTPRC+) and red blood cells (RBC, HBQ1+). Each cell cluster was composed of multiple fetal samples, and the samples of similar stages, or different sexes, were largely mixedly distributed (Fig. 1b, Supplementary Fig. 1d, e and 2a).

We analyzed the timing of endocrine cell differentiation. The results showed that the corticotropes appeared first at 7 weeks, the earliest time point we analyzed; this was immediately followed by the somatotropes and the gonadotropes at 8 weeks, and the thyrotropes and lactotropes appeared later at approximately 10 weeks and 16 weeks, respectively (Fig. 1d). The timing of endocrine cell differentiation was validated by immunofluorescence staining for the hormone genes, which was consistent with previous studies (Supplementary Fig. 2b)[15].

We analyzed a number of TFs that are mutated in human pituitary genetic diseases[16], showing that these TFs were expressed in a cell type-specific manner (Supplementary Fig. 1h). The SCENIC analysis identified activation of known TFs including SOX2, TBX19, POU1F1 and NR5A1 (Fig. 1e and Supplementary Fig. 2c)[17]. Taken together, these results indicated that our data provided comprehensive and precise information on human pituitary development.

**Characterizing pituitary stem cells.** Human pituitary stem cells have not been comprehensively characterized. We identified stem cell-specific genes including SOX2, PROP1, LHX3, HES1, ZFP36L1, ANXA1, NFIB, ZNF521 and NR2F2 (Fig. 2a). Gene set enrichment analysis (GSEA) showed that the TGF-β, Notch, Wnt and Hedgehog signaling pathways were enriched in the stem cells, and the "tight junction", "cell cycle" and "ECM receptor interaction" pathways were also highly enriched (Fig. 2b). We identified potential ligand-receptor genes between stem cells and mesenchymal cells, which enriched Gene Ontology (GO) terms such as "ECM receptor interaction" and the Notch, Wnt, BMP and Eph signaling pathways, suggesting that the mesenchymal cells may involve in regulation of the stem cells (Supplementary Fig. 3a, b and Supplementary Data 1). Co-immunofluorescence staining for SOX2 and Collagen III (COL3) suggested that mesenchymal cells encompass stem cells in human fetal pituitaries (Supplementary Fig. 3c).

Notably, reclustering the stem cells by Seurat revealed three subpopulations (Stem1, Stem2 and Stem3, Supplement Fig. 3d). Interestingly, Stem1 cells were found to be mainly derived from the early-stage pituitaries (7 to 10 weeks), while Stem3 cells were mainly derived from the late-stage pituitaries (19 to 25 weeks), indicating that these subpopulations represented time-dependent cell state shifts in the stem cells (Fig. 2c). The clustering result remained similar after regressing out the cell cycle genes (Supplement Fig. 3d).

Examination of differentially expressed genes (DEGs) among the three subpopulations identified 114 and 165 genes that were highly expressed in Stem1 and Stem3, respectively (Supplementary Fig. 3e). Stem1 cells were enriched in genes for "stem cell proliferation" (e.g. HMGA2), while the Stem3 cells were strongly enriched in genes for "negative regulation of cell proliferation" (e.g., CDKN1A, Fig. 2d and Supplementary Fig. 3e). Reclustering

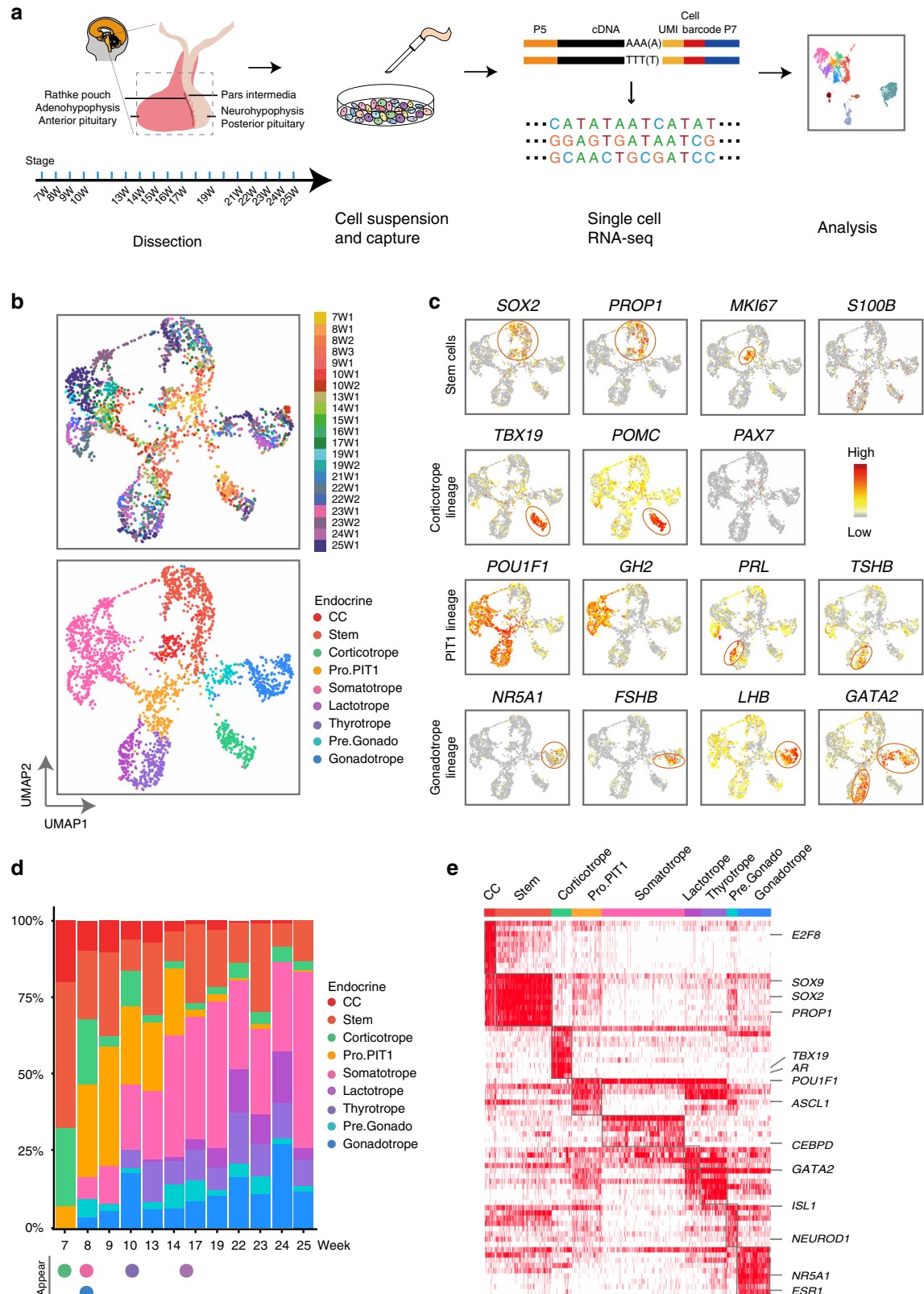

**Fig. 1 Diversity of cell types in the human fetal pituitary. a** Experimental flowchart for the scRNA-seq analysis of the human fetal pituitary. **b** UMAP plots showing the clusters of the cell cycle cells (CC), stem cells, the progenitor cells of PIT1 lineage (Pro.PIT1) or precursor cells of gonadotrope (Pre.Gonado) and the terminal endocrine cells (lower), and distribution of the fetal samples (upper). Dots: single cells. **c** Scatterplots showing expression of known markers, including TFs and hormone genes, projecting on the UMAP plot (b). Gray to red indicates no expression to high expression levels. **d** Bar plots showing the proportions of each cell type in each stage. Solid circles at the bottom indicate the earliest stages when a hormone producing cell type appears. **e** Heatmap showing the activated TFs predicated by SCENIC. For each cell type, the top 10 -log(P value) specific TFs being activated are shown, which ranked by number of cells. Columns are individual cells and rows are individual genes. White: not activated; red: activated. The expression levels of these TFs are shown in Supplementary Fig. 2c.

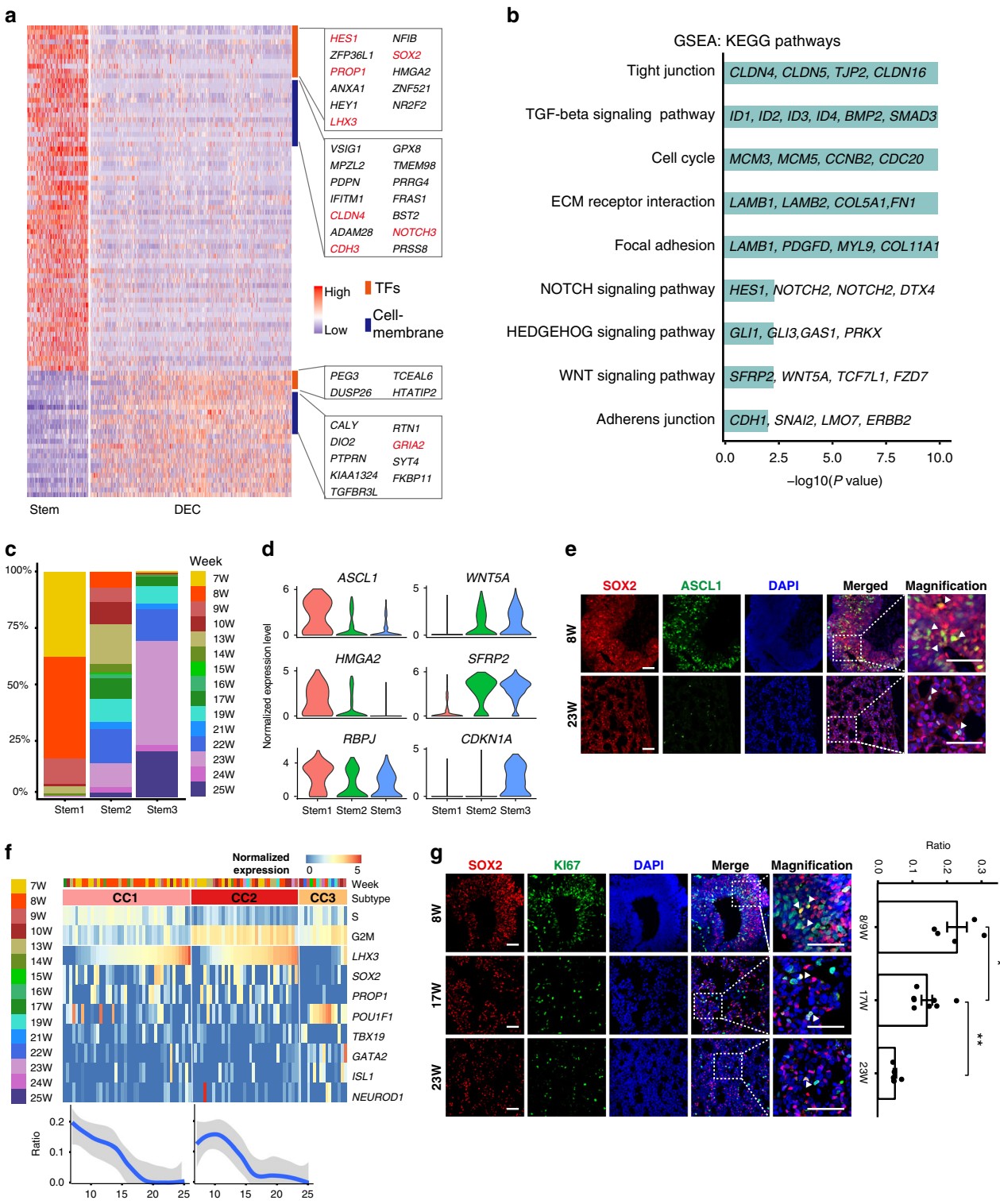

of the CC cluster showed that most cycling cells were *SOX2*-positive stem cells, and a small portion were *POU1F1* or *TBX19*-positive cells, which were verfied by immunofluorescence staining (Fig. 2f and Supplementary Fig. 3f). The proportion of cycling stem cells in the total stem cell population gradually decreased from the early to late stages (20% in the early stages, 10% in the middle stages, and less than 5% in the late stages; Fig. 2f), which was validated by coimmunostaining for SOX2 and Ki67 in the pituitaries of 8-, 17- and 23-week fetuses (Fig. 2g). These results

together suggest that Stem3 cells enter a quiescent or lowly proliferative state.

Interestingly, *ASCL1* was specifically expressed in Stem1 (Fig. 2d). This expression was consistent with the expression of *Ascl1* in the early stage of pituitary development in mouse and zebrafish[18,19]. We performed immunofluorescence staining to show that a large portion (24%) of SOX2-positive stem cells were also positive for ASCL1 at 10 weeks, while there were nearly no double-positive cells at 23 weeks (Fig. 2e). *ASCL1* is required for

**Fig. 2 Molecular characteristics and heterogeneity of pituitary stem cells. a** DEGs between the stem cells and the differentiated endocrine cells shown by a z-scored heatmap. DEC: differentiated endocrine cells, which include the progenitor or precursor cells. **b** Bar plots showing differentially enriched KEGG pathways between the stem cells and the DEC detected by GSEA. Representative genes in each pathway are shown. Nominal P values are determined by two-sided Kolmogorov-Smirnov Test and adjusted by FDR. **c** Bar plots showing the stages of three subtypes of the stem cells. The Stem1 are mainly derived from the early stages (7 to 10 weeks), and the Stem3 cells are mainly derived from the late stages (19 to 25 weeks). Color: weeks postfertilization. **d** Violin plots of representative DEGs between the Stem1 and Stem3 cells. **e** Immunofluorescence staining for SOX2 and ASCL1 in the 8- and 23-week human fetal pituitaries. Triangles indicate representative cells coexpressing both genes. Scale bar, 50 μm. **f** The CC cluster are composed of different cell subtypes. The upper panel shows the average expression levels of the S and G2/M phase genes, and representative cell type markers. CC1 and CC2 are SOX2-positive stem cells with CC1 and CC2 being the S and G2/M phase, respectively. CC3 are non-stem cells expressing POU1F1, TBX19 or GATA2. The lower panel shows the proportions of the cycling to all stem cells as a function of weeks postfertilization. Data are presented as mean ± SEM. **g** Immunofluorescence staining of SOX2 and Ki67 in the 8-, 17- and 23-week human fetal pituitaries (left). Triangles indicate representative costaining cells. The right panel shows the ratios of double-positive and SOX2-positive cells in each sample ($n \geq 2$). $p$-Value is determined by two-sided Wilcoxon test (*$P = 0.023$ which is less than 0.05, **$P = 0.0024$ which is less than 0.01). Data are presented as mean ± SEM. Scale bar, 50 μm.

differentiation of all anterior pituitary endocrine cell types in zebrafish, while in mice, mutation of *Ascl1* affects corticotropes and gonadotropes, and maybe also thyrotropes, at different regulatory steps[19–21]. The downregulation of *ASCL1* expression in the late pituitary stem cells may play a role in the state shift. The Wnt signaling modulator *SFRP2* and the non-canonical WNT gene *WNT5A* was prominently upregulated in Stem2 and Stem3 (Fig. 2d), which was consistent with the enrichment of the GO term "negative regulation of canonical Wnt signaling pathway" in Stem3 (Supplementary Fig. 3c).

The marker of the folliculostellate cells (FSCs) *S100B* was expressed in a small fraction (3.4%) of the stem cells of both the early and late stages (Fig. 1c). Comparison between the *S100B*-postive and *S100B*-negative stem cells revealed no DEGs relating to FSCs. The results suggested that FSCs were not present in the human fetal pituitary during 25 weeks postfertilization, which was consistent with previous immunostaining studies showing that S100-positive FSCs are basically not detected in the prenatal rat pituitary[22].

In summary, we identified an embryonic state shift of the pituitary stem cells accompanied by negative regulation of cell proliferation and alternations in TFs and signaling pathways.

**Pituitary stem cells in hybrid epithelial/mesenchymal state.** Previous studies have suggested that mouse pituitary stem cells undergo an epithelial-mesenchymal transition (EMT)-like process during differentiation[23,24]. We explored EMT dynamics during differentiation of the human pituitary stem cells. Among the Kyoto Encyclopedia of Genes and Genomes (KEGG) terms enriched in the stem cells, "tight junction" is related to the epithelial state, and "ECM receptor signaling pathways" is related to the mesenchymal state (Fig. 2b). By principal component analysis (PCA), all endocrine cells were ordered in a gradient transition from the stem cells to the differentiated cells on the principal component 1 (PC1) axis, so we used the PC1 axis as a trajectory for analyzing the stem cell differentiation (Fig. 3a). Examination of the expression of epithelial and mesenchymal markers showed that both the epithelial markers, such as *CDH1*, and the mesenchymal markers, such as *VIM* and *CDH2*, were highly expressed in the stem cells but lowly expressed in the differentiated cells; *EPCAM* was highly expressed in both groups (Fig. 3b and Supplementary Fig. 4b). This pattern suggested that the stem cells existed in a state expressing both the epithelial and mesenchymal markers, fitting the term of a hybrid E/M state as previously identified during mouse organogenesis and tumor transition[25,26].

To clarify the correlation between the hybrid E/M state and stemness, we defined an epithelial score (E.score), a mesenchymal score (M.score) and a stemness score (S.score) (Supplementary Data 2). We found that all three scores decreased along the

differentiation axis (Fig. 3c and Supplementary Fig. 4c). This pattern was exemplified by the expression levels of known stemness markers (*SOX2, SOX4, SOX9, NOTCH2* and *HES1*), epithelial markers (*EPCAM, CDH1, KRT8, CLDN4* and *GRHL2*) and mesenchymal markers (*CDH2, VIM, COL1A1, COL1A2* and *SNAI1*) which decreased along the timeline (Fig. 3c). We did not observe significant EMT changes in the stem cell substates (Supplementary Fig. 4d). The stem cells also specifically expressed *CDH3*, which plays roles in both maintaining stemness and promoting the hybrid E/M state in development and cancers (Fig. 2a)[27].

The SCENIC analysis suggested that *CDH1, CDH2, CDH3* and *VIM* were potential targets of *SOX2* and *SOX4*; *VIM* and *CDH3* were potential targets of *PROP1* whose targeted genes enriched in the "epithelial to mesenchymal transition" term, being consistent with the previous chromatin immunoprecipitation sequencing (ChIP-seq) study[24] (Supplementary Fig. 4e, f).

In sum, the data suggested that the pituitary stem cells existed in a hybrid E/M state which is associated with their stemness characters.

**Reconstructing developmental trajectories.** Next, we reconstructed the developmental progression of five endocrine cell lineages and identified transient precursors by applying the RNA velocity[28] and Slingshot[29] for the pseudotime trajectory analysis (Fig. 4a). The Slingshot results revealed lineage-shared and specific TFs being downregulated or upregulated during the pseudotime developmental progression, with some TFs showing peak expression in intermediated stages (Fig. 4b, Supplementary Data 3). A group of 29 TFs were identified as downregulated genes being shared among all five lineages, including *PROP1, SOX2, LHX3, HES1, TCF7L1* and *TGIF1* (Fig. 4c, d). Mutations in *TCF7L1* and *TGIF1* have been reported in patients with hypopituitarism recently[30,31]. *POU1F1, TBX19* and *NR5A1* were strongly upregulated in the PIT-1, corticotrope and gonadotrope lineages, respectively. *GATA2, FOXL2* and *ISL1* were shared by the thyrotrope and gonadotrope lineages. *NEUROD1* showed peak expression at the intermediated states of both the gonadotrope and corticotrope lineages (Fig. 4c).

**Corticotropes experience two subtates.** Next, we investigated development of each cell lineage. The corticotrope is the first hormone-producing cell type that appears and *TBX19* is crucial for its development[3]. The pseudotime and reclustering analysis identified two subclusters: Corticotrope1, which comprised cells mainly derived from the early stage (7 to 9 weeks), and Corticotrope2, which comprised cells mainly derived from the middle or late stages (10 to 25 weeks) (Fig. 5a). *TBX19* and *POMC* were highly expressed in both clusters, which was consistent with the

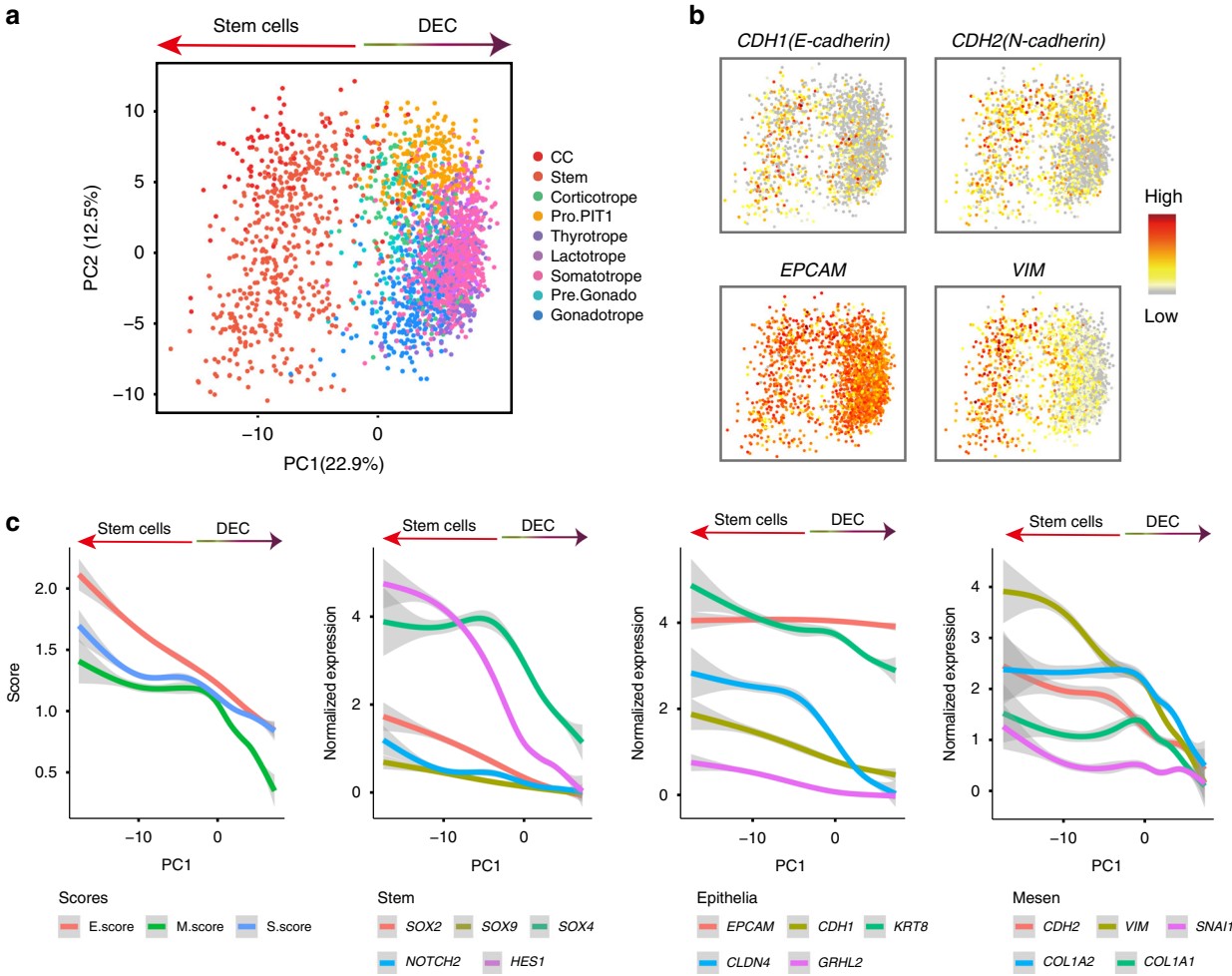

**Fig. 3 Hybrid epithelial/mesenchymal state of pituitary stem cells. a** PCA plot showing differentiation of endocrine cells along the PC1 axis. Dots: single cells; colors: cell types. **b** Scatterplots showing the expression of representative epithelial and mesenchymal markers projecting on the PCA plot. Gray to red indicate no to high expression levels. **c** The values of the epithelial score (E.score), the mesenchymal score (M.score), and the stemness score (S.score), as well as and the expression levels of representative genes, decreased during differentiation. The colored lines represent loess-smoothed expression. Data are presented as mean ± SEM.

previous immunofluorescence staining study showing that all *TBX19*-positive cells also express *POMC* (Fig. 5b, c)[32]. Androgen receptor (*AR*) was prominently expressed in both corticotrope clusters, which was predicted to be a potential TF regulating *POMC* by SCENIC, and the expression was verified by immunofluorescence (Figs. 1e, 5b and c). AR mutant mice have been shown to exhibit an increased expression of *POMC*[33].

We identified DEGs between Corticotrope1 and Corticotrope2. DEGs in Corticotrope1 enriched GO terms such as "regulation of mitotic cell cycle" (e.g., *HMGA2*), suggesting that they could potentially proliferate (Fig. 5d, e, and Supplementary Data 4). Indeed, a few *TBX19*-positive cycling cells were captured (Fig. 2g). Interestingly, follistatin (*FST*), which is an antagonist of ACTVIN, was specifically expressed in Corticotrope1, suggesting that this intermediate cell type may also play some paracrine roles for regulating the gonadotropes. DEGs in Corticotrope2 enriched GO terms such as "response to corticosterone" (e.g., *NR4A1* and *NR4A2*), suggesting that these cells were more matured (Fig. 5d, e)[34].

Another *POMC*-expressing lineage is the melanotrope in the intermediate lobe, which appears after 14 weeks of gestation but is scarce in the human pituitary[15]. The melanotrope coexpresses *TBX19* and *PAX7*, both of which are required for its differentiation[35]. Only 4 cells, which were 7- or 8-week stem cells,

coexpressed *TBX19* and *PAX7* in our data (Fig. 1c). Comparison between these cells and other stem cells revealed only *PAX7* and three other DEGs that were not expressed in the mouse and rat melanotropes. These results suggested that we have not captured the melanotrope.

In summary, we defined corticotrope differentiation from an early intermediate state to a maturing state with upregulation of the expression of genes to establish the cortisol feedback.

**PIT-1 lineages segregate from a common progenitor.** The PIT-1 lineage is comprised of three endocrine cell types: the somatotrope, the lactotrope and the thyrotrope, all of which are governed by *POU1F1*[2,36]. In addition to these three hormone producing cell types, the pseudotime trajectories analysis identified three intermediated progenitor or precursor cell populations: the Pro.PIT1_all cells as a common progenitor for all three hormone producing cell types (146 cells), the Pre.Thy as a precursor for the thyrotrope (74 cells, *GATA2*-positive) and the Pre.Som as a potential precursor for the somatotrope (19 cells, Fig. 6a, b).

To explore how three different lineages segregate from the common progenitor cell state, we first investigated expression dynamics of known lineage-enriched genes (Fig. 6c). Previous studies have shown that mutation of *Neurod4* in mice leads to almost completely lack of *GHRHR* expression and markedly

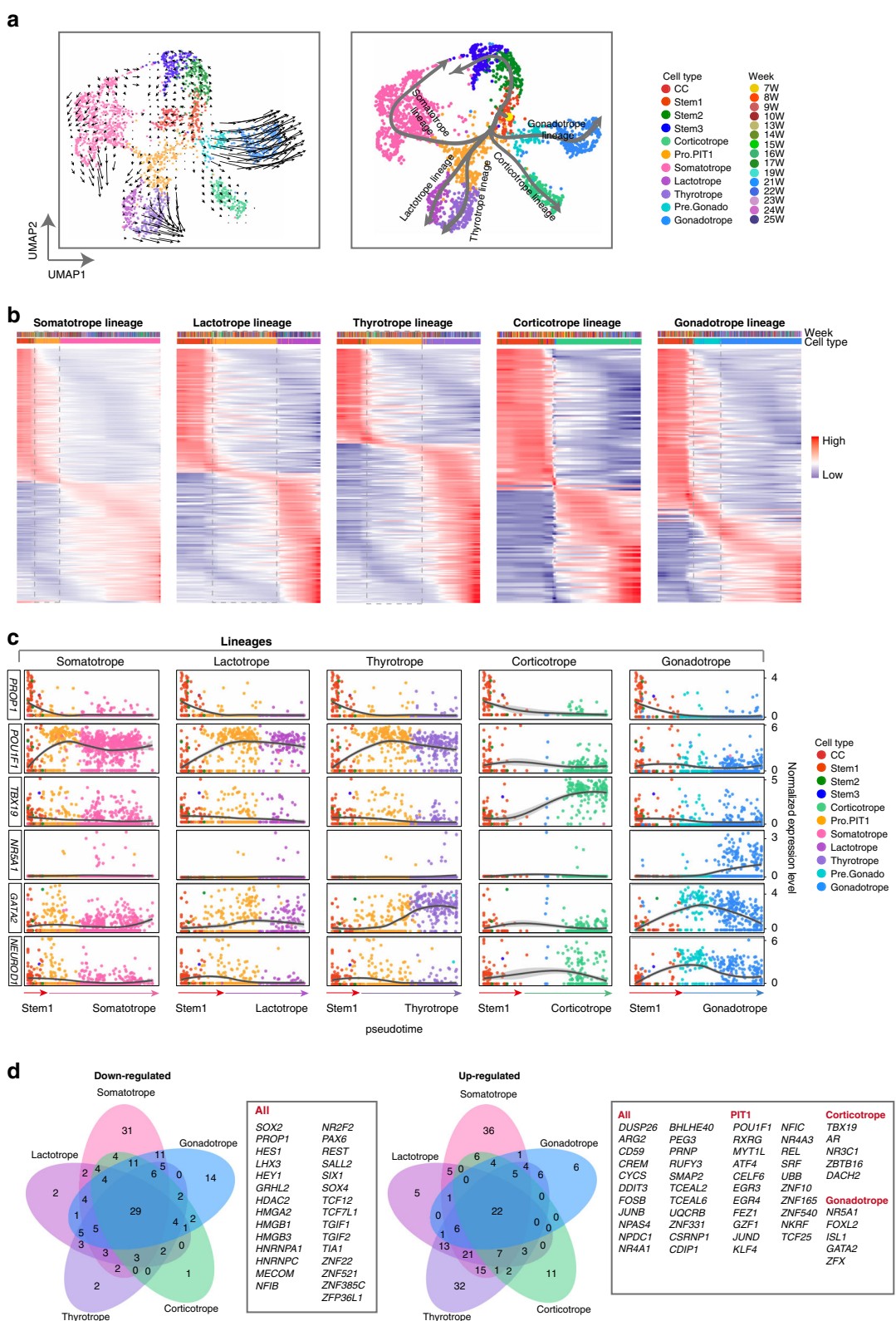

decrease in *GH* expression[21,37]. Mutation of *Foxo1* in mice also results in delayed somatotrope differentiation[38]. Notably, we found that *NEUROD4* was clearly activated in Pro.PIT1_all cells in comparison with the stem cells (logFC = 1.6, *P* = 1.3E–18, Fig. 6c). *NEUROD4* was further significantly upregulated in the Pre.Som cells and the differentiated somatotropes. *FOXO1* was

not expressed in the Pro.PIT1_all cells before upregulation in the Pre.Som cells and a portion of the somatotropes. Thus, the results demonstrated that *NEUROD4* played roles earlier than *FOXO1* in the somatotropes.

*ZBTB20* has been recently demonstrated to be crucial for lactotrope specification in mice[39,40]. This gene was expressed

**Fig. 4 Pseudotime developmental trajectories of hormone-producing cells. a** Pseudotime analyses of the endocrine cells shown in the UMAP plot. Left: the RNA velocity result with the arrows predicting directions of the pseudotime. Right: the Slingshot result with the lines indicating the trajectories of lineages and the arrows indicating manually added directions of the pseudotime. Dots: single cells; colors: cell types. Yellow circle in the right panel represents the start point of the trajectories which was set as the Stem1 subcluster. Since the corticotrope was already a separate cluster in the earliest sample (7 weeks), the corticotrope trajectory may start from stem cells earlier than Stem1. The CC cluster was omitted in the Slingshot pseudotime analysis as it contained several cell types and caused wrong trajectories. **b** Heatmap showing the relative expression of TFs displaying significant changes ($P \leq 1E-5$) along the pseudotime axis of each lineage. The progenitor or precursor cells of each lineage were enclosed in frames of dashed lines. Colors: loess-smoothed expression (red, high; blue, low). The columns represent the cells being ordered along the pseudotime axis with the cell type and fetal week informations being shown above. Rows represent genes being ordered by their peak expression along the pseudotime axis. $P$ values are determined by one way ANOVA test. **c** Scatterplots showing the expression levels of representative TFs along the pseudotime axis. Dots: single cells; colors: cell types. **d** Venn diagram of downregulated and upregulated TFs along the pseudotime axis of each lineage. Representative TFs were listed in the right boxes.

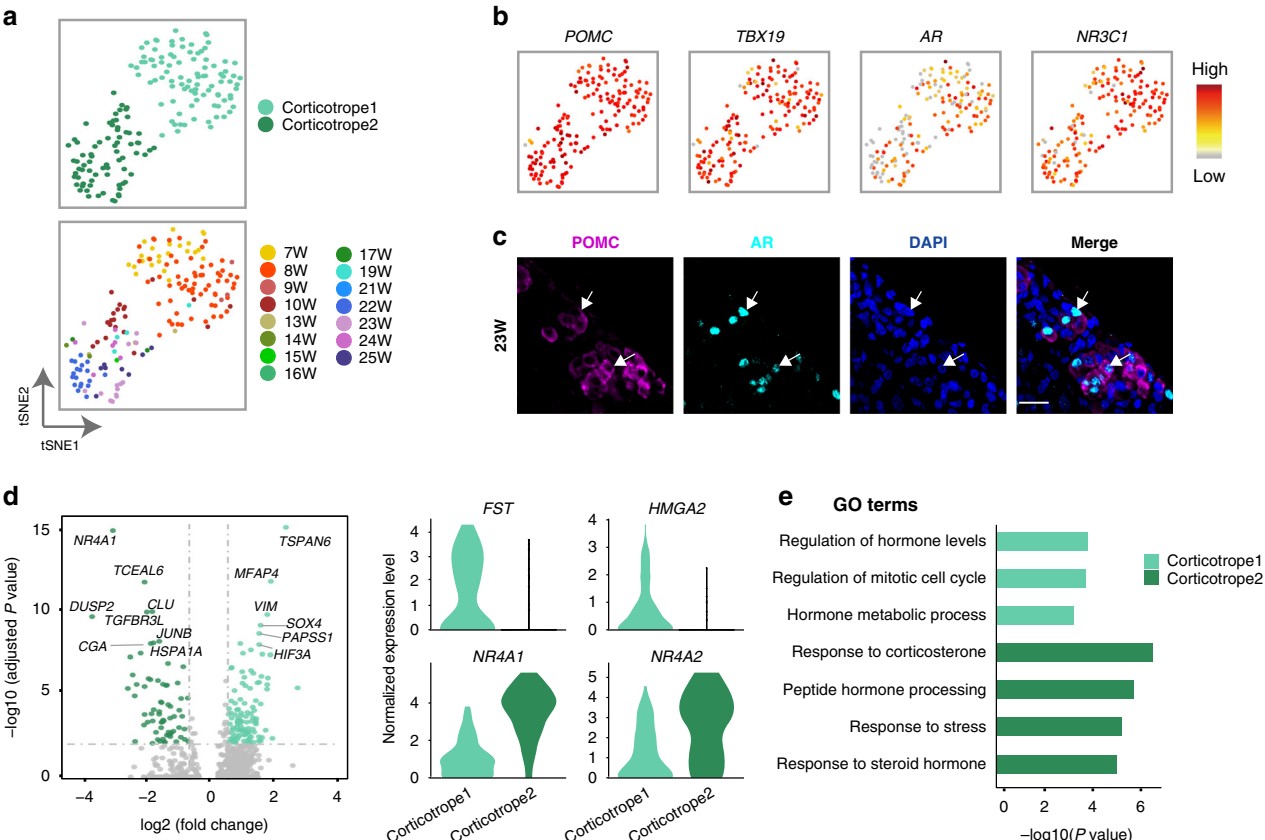

**Fig. 5 Characteristics of two corticotrope substates. a** *t*-SNE plots of corticotropes of different subtypes (upper) and fetal stages (lower). Dots: single cells; colors: cell types or weeks postfertilization. **b** Scatterplots showing expression of representative genes (*POMC*, *TBX19*, *AR* and *NR3C1*). Gray to red indicate no to high expression levels. **c** Immunofluorescence staining of POMC and AR (23 weeks). Scale bar, 20 μm. **d** Volcano (left) and violin (right) plots showing DEGs between Corticotrope1 and Corticotrope2. Colored dots indicate significant DEGs in each subtype. *P*-values are determined by two-sided Wilcoxon test and adjusted by Bonferroni correction. **e** Bar plots showing GO terms being enriched in Corticotrope1 and Corticotrope2. *P*-values are determined by one-sided test.

in all pituitary cell types including the stem cells. Notably, its expression level was significantly upregulated in the Pro.PIT1_all cells in comparison with the stem cells (logFC = 0.6, $P = 9.8E-08$), and the level was similar between the Pro.PIT1_all cells and the lactotropes ($P > 0.1$, Fig. 6c). Due to downregulation in other two lineages, the expression level of *ZBTB20* was slightly but significantly higher in the lactotropes comparing with the somatotropes and the thyrotropes (Lactotrope versus Somatotrope: logFC = 0.7, $P = 9E-25$; Lactotrope versus Thyrotrope: logFC = 0.4, $P = 1.1E-08$). This expression pattern was consistent with the essential role of *ZBTB20* in lactotrope specification.

*GATA2*, and possible *ASCL1*, have been implicated in thyrotrope development in mouse[18,21,41]. In zebrafish, *sox4b*

has been shown to be required for thyrotrope development by activating *gata2a* expression[42]. We found that *ASCL1* and *SOX4* were prominently expressed in the Pro.PIT1_all cells. *GATA2* was not significantly upregulated in the Pro.PIT1_all cells ($P > 0.1$), but were prominently activated in the Pre.Thy and further upregulated in the terminal differentiated thyrotropes. Interestingly, *SOX11*, a member of SoxC family genes with *SOX4*, was significantly upregulated in the Pro.PIT1_all cells in comparison with the stem cells (logFC = 1.0, $P = 4.2E-12$, Fig. 6d). The SCENIC analysis also suggested that both SOX4 and SOX11 bind to the regulatory region of *GATA2*. *ASCL1* displayed significantly higher expression levels in the thyrotropes than two other lineages (Thyrotrope versus

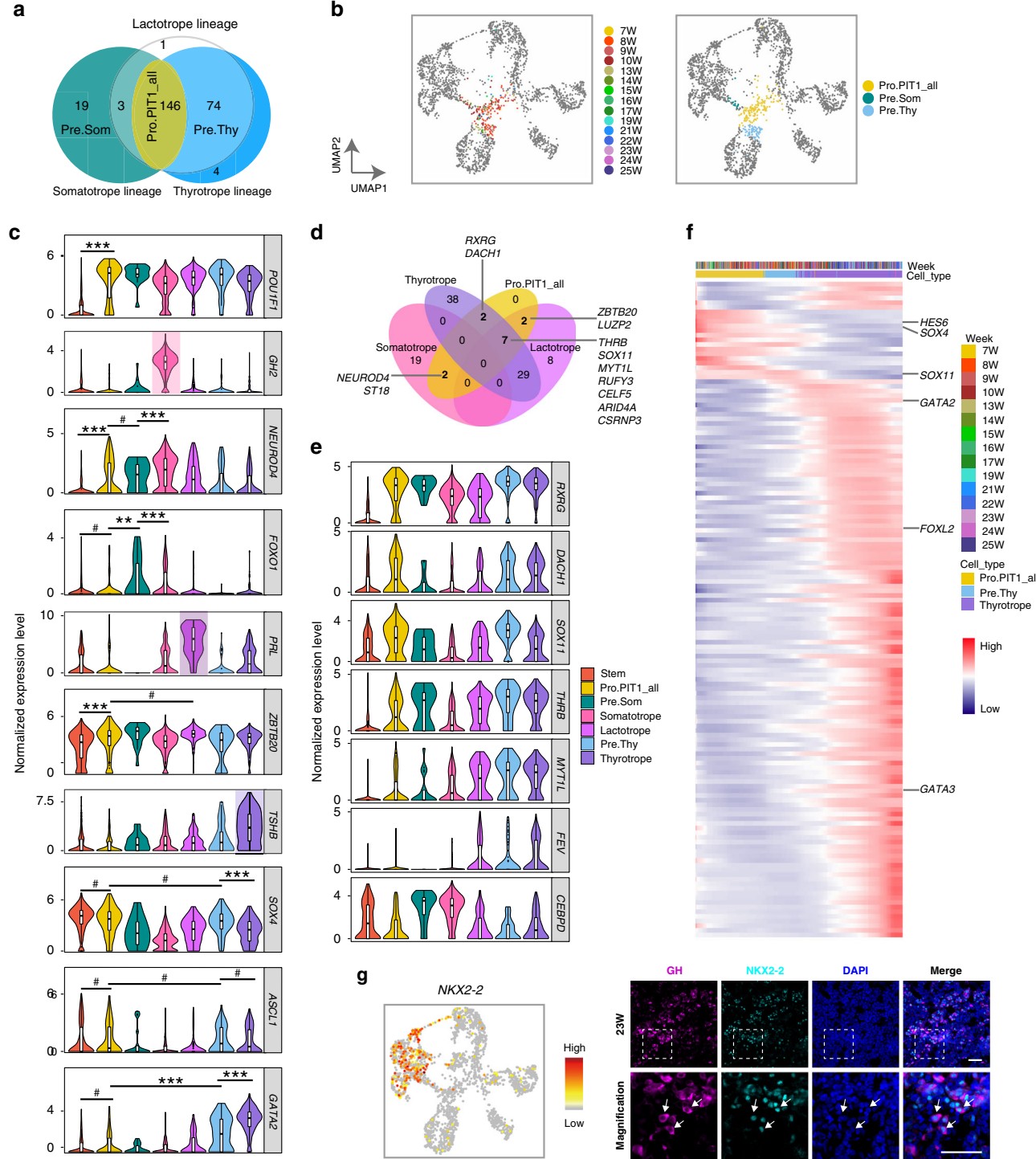

**Fig. 6 Differentiation trajectories and TF dynamics of the PIT-1 Lineage. a** Venn diagram of progenitor cells distributed in each PIT1 sublineage as shown in the pseudotime analysis. Pro.PIT1_all: the common progenitor cells of all three sublineages; Pre.Thy, the precursor cells of thyrotropes; Pre,Som: the potential precursor cells of somatotropes. **b** Distribution of the subtypes of the Pro.PIT1 on the UMAP plot. Dots: single cells; colors: weeks postfertilization (left) or cell types (right). **c** Violin and inside box plots showing the gene expression dynamics of known TFs for each PIT1 lineage. #*P* > 0.1, **P* < 0.01 ***P* < 0.001. *P*-values are determined by two-sided Wilcoxon test and adjusted by Bonferroni correction. **d** Venn diagram showing the intersections among the differentially expressed TFs between any two of the three PIT1 lineages, and the 13 TFs that were significantly upregulated in the Pro.PIT1_all cells in comparison with the stem cells, for which the names of genes are shown. **e** Violin with inside box plots showing representative identified TFs for each PIT1 lineage. **f** Heatmap showing the relative expression levels of the significantly upregulated TFs of thyrotropes along the pseudotime axis. Colors: loess-smoothed expression (red, high; blue, low). The columns represent cells being ordered along the pseudotime axis, and cell type information is shown above the heatmap. Rows represent genes being ordered by their peak expression along the pseudotime axis. **g** The scatterplot (left) expression and immunofluorescence staining of *NKX2-2* as a somatotrope-specific TF. Scale bar, 50μm. Arrows indicate representative cells co-expressing GH and NKX2-2.

Somatotrope: logFC = 1.3, $P$ = 6.6E-15; Thyrotrope versus Lactotrope: logFC = 0.7, $P$ = 3.7E-09).

Then, we investigated a comprehensive view of lineage-enriched genes and identified a total of 1,277 DEGs, including 107 TFs, between any two of three lineages (logFC ≥ 0.5 and adjusted $P$ ≤ 0.01, Fig. 6d, Supplementary Fig. 5a, b and Supplementary Data 5). Thirteen TFs were significantly upregulated in the Pro.PIT1_all cells comparing with the stem cells. Among these TF, remarkably, *NEUROD4* and *ZBTB20* were identified as the primary lineage-enriched TFs for the somatotropes and the lactotropes, respectively (Fig. 6c, d). *RXRG* and *DACH1* were identified as two thyrotrope-enriched genes, both of which were prominently upregulated in the Pro.PIT1_all cells and kept the high expression levels in the thyrotropes while downregulated in the somatotropes and the lactotropes (Fig. 6c, d).

A total of 19 somatotrope-enriched TFs, including *FOXO1*, *CEBPD* and *NKX2-2*, were upregulated during the differentiating process of the somatotrope. *NKX2-2*, which has been shown to be essential for development of neuroendocrine, gastrointestinal tract and pancreas[43], was specifically upregulated in a portion of terminal somatotropes (Fig. 6g), and it is highly expressed in the human adult pituitary in the GTEx database. We validated the coexpression of NKX2-2 with GH by immunofluorescence (Fig. 6g). A total of 8 lactotrope-enriched TFs (e.g. *SIX6*) and 38 thyrotrope-enriched (e.g., *GATA2*, *ISL1* and *FOXL2*) were upregulated during the differentiation of the lactotropes and the thyrotropes, respectively (Fig. 6d, e). The pseudotime analysis of thyrotrope-lineage differentiation showed that expression of *SOX4* and *SOX11* peaked before activation of *GATA2* in the Pre.Thy cells (Fig. 6f). *SOX4* and *SOX11* were significantly downregulated in the thyrotrope comparing with the Pre.Thy (*SOX4*: logFC = 1.0, $P$ = 2.7E-7; *SOX11*: logFC = 1.3, $P$ = 4.9E-13).

A surprising finding was that the lactotropes and the thyrotropes were close to each other relative to the somatotropes. This was displayed in UMAP, and fewer DEGs were identified between these two lineages comparing with the somatotropes, and the transcriptomes of the lactotropes and the thyrotropes were closely correlated (Figs. 4a, 6d and Supplementary Fig. 5a, b, c). In contrast, the somatotropes and lactotropes were more close to each other comparing with the thyrotropes in the mouse and rate adult pituitaries (Supplementary Fig. 5c). We speculated that this may be due to that both the lactotropes and the thyrotropes were in less matured states comparing with the somatotropes before 25 weeks. In the human fetus, GH begins to secrete before 10 weeks and peaks at approximate 22 weeks, while PRL begins to secrete at 25 weeks and peaks at birth[44]. Supporting this, we found that *TRHR* and *ESR1* were lowly expressed in the fetal thyrotropes and lactotropes, respectively.

Furthermore, despite of many evidences suggesting the existence of mammosomatotropes coexpressing GH and PRL in the human and rodent pituitary[45,46], our analysis did not identify a distinct cell cluster that corresponds to a common precursor of the somatotropes and the lactotropes, even after removing potential batch effects of the somatotropes and the lactotropes; the result was similar to that of the rat scRNA-seq study[9] (Supplementary Fig. 5d).

Taken together, our results characterized the transcriptome dynamics during specification of the PIT-1 lineages, in which a common progenitor coexpresses lineage-enriched TFs prior to activation of divergent TF networks.

**Gonadotropes exhibit two developmental trajectories.** Gonadotropes mainly secrete two types of hormones, LH and FSH, which are essential for reproduction in both sexes.

The pseudotime and reclustering analyses interestingly identified five subclusters (Fig. 7a). There was a clear intermediate precursor cell state, Pre.Gonado, which expressed *GATA2* and *FOXL2*, but not *NR5A1*. The other four clusters (Gonadotrope1, 2, 3 and 4) comprised cells that expressed *NR5A1* and *GNRHR* with different expression patterns of *LHB*, *FSHB* and the primate-specific hormone chorionic gonadotropin (CGBs). *NR5A1, GNRHR* and *LHB* were expressed in the four cell clusters at a similar level, while *CGBs* were expressed solely in Gonadotrope2 ($LHB^{high}CGB^{high}FSHB^{low}$), and *FSHB* was more highly expressed in Gonadotrope4 ($LHB^{high}CGB^{low}FSHB^{high}$).

The pseudotime analysis revealed two developmental trajectories: one trajectory was from the Pre.Gonado to the Gonadotrope1 and terminated at the Gonadotrope2 (Type I trajectory), while the other trajectory was from the Pre.Gonado to the Gonadotrope3 and Gonadotrope4 (Type II trajectory, Fig. 7a, b). For the Type I trajectory, the intermediate Gonadotrope1 was solely in the early stages (8 to 14 weeks) while Gonadotrope2 was comprised of both early and late stages; all Type II trajectory cells were in the late stages (15 to 25 weeks). These results indicated that the Type I and II trajectories represented an early and a late gonadotrope lineage, respectively. Among all five clusters, Gonadotrope2 had the most DEGs, and comparing between Gonadotrope2 and Gonadotrope4 identified 265 and 30 DEGs, respectively (Fig. 7c, Supplementary Fig. 6 and Supplementary Data 6). GO analysis showed that Gonadotrope2 DEGs enriched "regulated exocytosis" and "C21 − steroid hormone biosynthetic process" (e.g. the steroidogenic enzyme *CYP11A1*), suggesting that Gonadotrope2 actively secreted hormones (Fig. 7c). *WNT4* and *GATA2* were more highly expressed in Gonadotrope2 and Gonadotrope4, respectively (Fig. 7d). Other DEGs included *HIF3A* and *MC2R* in the Type I trajectory and folate receptor *FOLR1* and secretoglobin *SCGB2A1* in the Type II trajectory (Fig. 7d and Supplementary Fig. 6).

Together, these data determined two gonadotrope developmental trajectories including a previously unappreciated $LHB^{high}CGB^{high}FSHB^{low}$ subcluster.

**Species comparison between human and rodent pituitaries.** Next, we compared our scRNA-seq data with two recently published scRNA-seq datasets of mouse and rat adult pituitaries[9,10]. The human scRNA-seq data, which used a plate-based method, recovered higher number of genes per cell comparing with the rodent data using the droplet-based 10X genomic method (Supplementary Fig. 7a). Integrating three datasets showed that most pituitary cell types were conserved among human and rodent (Fig. 8a). Anterior pituitary known markers and new markers, including *SOX2, POMC, GH1, PRL, TSHB, GNRHR, ALDH1A2, NR3C1, DLK1, OLFM1, DIO2* and *KCNK3*, showed similar cell-type-specific expression patterns (Fig. 8b, Supplementary Fig. 7b).

Notably, no progenitor or precursor cell types were found in the rodent datasets, indicating that differentiation rarely occurs in these adult pituitaries. Also, in the human fetal pituitaries, the proliferating cells were mainly the stem cells, while in the adult rodent pituitaries, the proliferating cells were mainly the somatotropes and the lactotropes. The rat FSCs were clustered closely to human and mouse stem cells; all expressed *SOX2*, but only the FSCs expressed *S100B*.

We then attempted to find species-specific genes, despite that any differences between the fetal human dataset and the adult rodent datasets could reflect the species or stage differences. We recognized cell-type-specific genes for each species and then identified human (or fetal)-specific genes and rodent (or adult)-specific genes (Supplementary Data 7). Among the identified genes, *CGB* and *GH2* are primate-specific and thus do not exist in

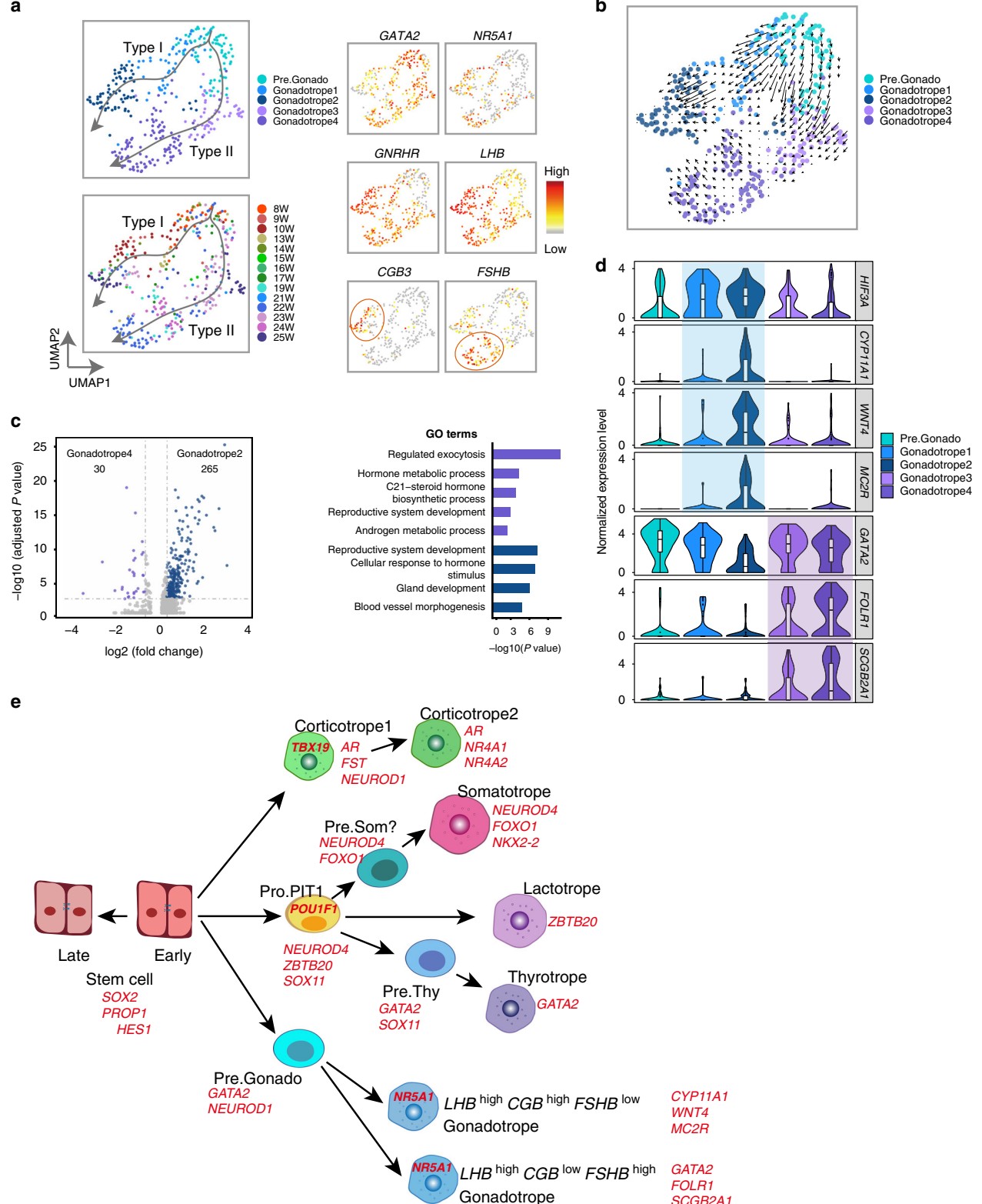

**Fig. 7 Two developmental trajectories of gonadotropes.** **a** UMAP plots (left) showing that Gonadotropes are comprised of two sublineages with scatterplots (right) showing the expression patterns of representative marker genes. Dots: single cells; colors: cell types (upper) or weeks postfertilization (lower). Gray to red indicate no to high expression levels. **b** RNA velocity analysis of gonadotropes projected onto the UMAP plot showing two sublineages. Dots: single cells; colors: cell types; arrows: predicted directions of the pseudotime. **c** Volcano plots (left) of DEGs between Gonadotrope2 and Gonadotrope4, with bar plots (right) showing GO terms enriched in each subtype. Colored dots indicate significant DEGs in each subtype. *P*-values are determined by two-sided Wilcoxon test and adjusted by Bonferroni correction. **d** Violin with inside box plots showing representative DEGs of two sublineages. **e** Diagram of the development of human fetal anterior pituitary. For a given stage, representative enriched TFs are highlighted. The Pre.Som is labeled with a question mark as this intermediated substate is supported by sufficient number of cells.

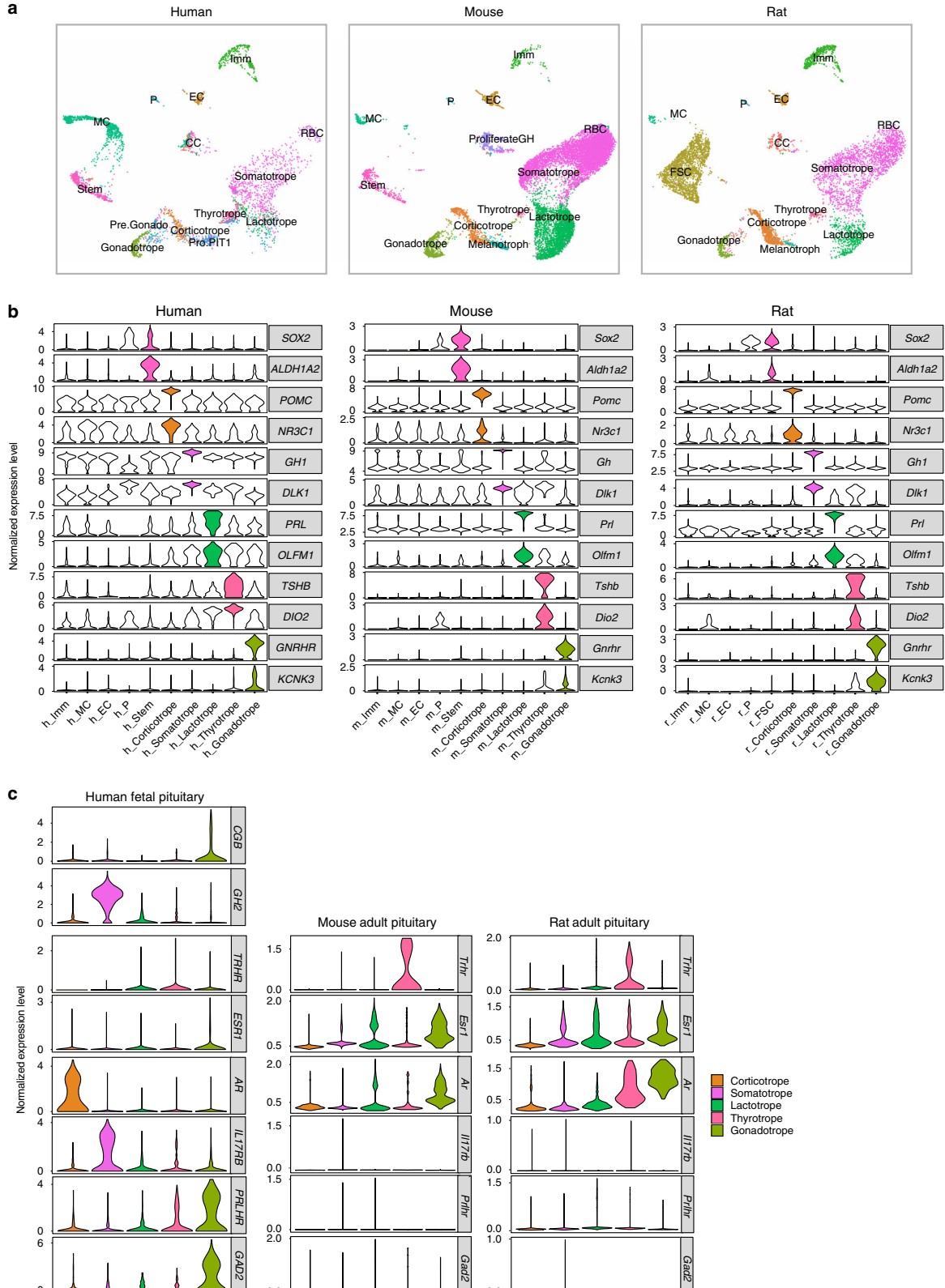

**Fig. 8 Comparison among human and rodent pituitaries. a** UMAP plots of the pituitary cell types by integrating human and rodent pituitary scRNA-seq data sets, with the cell types identified in each data set being shown seperatedly. P, pituicytes; MC, mesenchymal cells; EC, endothelial cells; Imm, immune cells; RBC, red blood cells. **b** Violin plots of the representative identified markers shared among human, mouse and rat. **c** Violin plots of the representative distinct cell type-specific genes between the human fetal pituitary and rodent adult pituitaries.

the rodent data (Fig. 8c). *Trhr* and *Esr1* were highly expressed in the rodent thyrotropes and lactotropes, respectively, but lowly expressed in the human fetal pituitary, thus reflecting stage differences. Interestingly, *AR* was highly expressed in the human fetal corticotropes while lowly expressed in the rodent adult corticotropes instead with high expression in the gonatropes (Fig. 8c). Also, *PRLHR* was highly expressed in the human fetal gonatropes while nearly not expressed in rodent[47]. *IL17RB* and *GAD2* also showed specific expression in the human fetal pituitary. Together, these results revealed a small number of potential species-specific genes.

## Discussion

We applied scRNA-seq to elucidate transcriptomic heterogeneity and dynamics in the human fetal pituitary. First, we characterized divergent developmental trajectories with distinct transitional intermediate states in five pituitary hormone-producing cell lineages (Figs. 4a and 7e). For the PIT-1 lineage, integrating our scRNA-seq data with previous mouse genetic studies provides insights into how three distinct cell types are specified from a common progenitor cell. We found that lineage-enriched TFs, including *NEUROD4, ZBTB20, ASCL1, SOX4* and *SOX11*, were coexpressed in the progenitor cells (*NEUROD4*$^{mid}$/*ZBTB20*$^{high}$/ *ASCL1*$^{high}$). During differentiation, the cells further segregate into the *NEUROD4*$^{high}$/*ZBTB20*$^{mid}$/*ASCL1*$^{low}$ somatotropes, the *NEUROD4*$^{mid}$/*ZBTB20*$^{high}$/*ASCL1*$^{low}$ lactotropes and the *NEU-ROD4*$^{low}$/*ZBTB20*$^{mid}$ /*ASCL1*$^{high}$ thyrotropes. This finding is consistent with the results of recent scRNA-seq studies demonstrating coactivation of alternative programs preceding two-cell fate commitment in several developmental systems[48–50]. A recent mouse genetic study interestingly showed that the lactotropes were significantly increased in the *Neurod4*-null mice, with a more prominently increase in the *Ascl1;Neurod4* double knockout mice[21]. Thus, it seems that there is competition among the three programs in the progenitor cells with the lactotrope lineage being produced in a situation of high *ZBTB20* expression without high activity of alternative lineage-enriched *NEUROD4* and *ASCL1*.

How thyrotropes are specified in mammals has not been sufficiently clarified. We found that two SoxC family genes, *SOX4* and *SOX11*, were prominently expressed in the Pro.PIT1_all cells before activation of *GATA2* and *GATA3* in the Pre.Thy (Fig. 6f), suggesting that the SOX genes play roles in thyrotrope commitment by regulating the GATA genes as sox4b does in zebrafish; these two SoxC genes may act redundantly as in other developmental processes[51]. Another interesting candidate gene is *DACH1*, which was thyrotrope-enriched and clearly upregulated in the progenitor cells. A previous study has suggested that *Six6* interacts with *Dach1* for regulating cell proliferation in pituitary; however, the function of *DACH1* on the thyrotrope development has not been addressed[52,53]. It should be noted that *DACH1* was also prominently expressed in Pre.Gonado.

The developmental trajectories of the corticotrope and gonadotrope lineages were different from the developmental trajectory of the PIT1 lineage. Corticotrope development was relatively straightforward. The terminal *POMC* gene was coexpressed with *TBX19* even in the early intermediate state. We did not capture a *TBX19*$^+$/*POMC*$^-$ state, suggesting that this state may be very transient or much earlier than cells we collected. In contrast, the gonadotrope lineage was characterized by two multistep developmental trajectories that clearly contained a *GATA2*$^+$/ *POU1F1*$^-$/*NR5A1*$^-$/*LHB*$^-$ intermediate state before terminal differentiation. Interestingly, an interaction between the corticotrope and the gonadotrope lineages seems to exist during the fetal stage. The *LHB*$^{high}$*CGB*$^{high}$*FSHB*$^{low}$ fetal gonadotrope subtype specifically expresses *MC2R*, which is the ACTH receptor. Additionally, the early

intermediate corticotrope subtype (Corticotrope1) specifically expressed *FST*, which functions as an activin antagonist to inhibit FSH release, and consistently, *FSHB* was expressed at low levels in this fetal gonadotrope subtype (Figs. 5d and 7a).

Second, we identified a fetal gonadotrope subtype (the *LHB*$^{high}$*CGB*$^{high}$*FSHB*$^{low}$ cells of the Type I trajectory). The hypothalamic-pituitary-gonadal (HPG) axis is activated in the mid-gestational human fetus[54]. Our data suggested that the *LHB*$^{high}$*CGB*$^{high}$*FSHB*$^{low}$ cells matured at 10 weeks postfertilization, and thus the developmental timing of these cells matches the activation of the fetal HPG axis. The development of the Type II trajectory occurs after 15 weeks postfertilization, lagging behind the Type I trajectory. In addition, our data suggested that the *LHB*$^{high}$*CGB*$^{high}$*FSHB*$^{low}$ cells are matured with a feature of actively secreting hormones. These results together indicate that the *LHB*$^{high}$*CGB*$^{high}$*FSHB*$^{low}$ cells play a role in establishing the early fetal HPG axis. The fetal HPG axis is essential for development of the male genitalia, yet other biological significances are not fully understood[55]. After mid-gestation, the HPG axis is silenced towards the end of gestation, and reactivated at birth (also called minipuberty), and suppressed throughout childhood until reactivation at puberty[56]. It is possible that the *LHB*$^{high}$*CGB*$^{high}$*FSHB*$^{low}$ cells are also involved in these regulations. Interestingly, it seems that this fetal gonadotrope subtype does not exist in mice since CGB are a primate-specific hormone and the gonadotrope cell type is the last cell type to reach maturation in mice[1]; in contrast, we found that this subtype is among the earliest generated cell types in human.

Third, we characterized a hybrid E/M state in the human fetal pituitary stem cells. Previous studies have suggested that pituitary stem cells undergo an EMT-like process for cell migration and differentiation[23,24,57,58]. Particularly, mice deficient of *Prop1* reveal impaired migration of stem cells, and *Prop1* has been shown to directly target both the epithelial and mesenchymal genes[24,58]. Our results showed that most typical EMT TFs are not expressed in the stem cells, except weak expression of *SNAI1*, *SNAI2* and *ZEB1*, indicating that the cells are not undergoing full EMT, and this is consistent with our previous study on mouse organogenesis[25]. It is noteworthy that the coexpression of the epithelial and mesenchymal markers is also detected in mouse adult stem cells and rat FSCs (Supplementary Fig. 7b). However, it is possible that, since we only analyzed limited number of cells from 7 weeks on, we have not captured all E/M substates of the pituitary stem cells including the early *SOX2*$^+$/*PROP1*$^-$ cells and other transitioning cells.

Collectively, this study provides key insights into the transcriptional landscape of human pituitary development, defining distinct cell substates and subtypes, and illustrating transcriptional implementation during major cell fate decisions. The data may also help identify disease genes of congenital hypopituitarism and provide a reference for human pluripotent stem cells-generated anterior pituitary tissue for therapeutic application and disease modeling[59].

## Methods

**Human fetal pituitary dissection**. The donors in this study were pregnant women who could not continue pregnancy because of their own diseases (such as cervical insufficiency, inevitable abortion, infection, eclampsia, as examples). All patients voluntarily donated the fetal tissues and signed informed consents. This study was approved by the Reproductive Study Ethics Committee of Peking University Third Hospital (2017SZ-043).

We collected 21 human fetal pituitaries from fetuses at 7 to 25 week postfertilization (corresponding to 9 to 27 weeks of gestation), including 11 female fetuses and 10 male fetuses. The pituitary tissues were dissected under a dissecting microscope. For fetuses earlier than 14 weeks, the whole pituitaries were analyzed; for fetuses later than 14 weeks, we separated the anterior and posterior pituitaries, and only analyzed the anterior pituitaries except two fetuses (15W1 and 17W1) of

which we collected cells from both parts. A total of 58 high-quality cells were obtained from the posterior pituitaries of these two fetuses, most of which ($n = 43$) were pituicytes and other cells included 12 immune cells and 3 red blood cells. The tissues were washed twice with Dulbecco's phosphate buffered saline (DPBS) and then the minced tissues were digested with 1 mg/ml type II (17101015, GIBCO) and type IV collagenase (17104019, GIBCO) at 37 °C for 15–30 min. After filtration through 40-µM nylon mesh, the cells were washed once with 10% FBS DMEM, and a single-cell suspension was obtained.

**Immunohistochemistry and immunofluorescence assays**. After the whole pituitary was dissected and washed three times with DPBS, the tissue was fixed with 4% paraformaldehyde at 4 °C overnight.

For histological analysis, fixed tissue was stained with H&E. For IF, after washing and dehydration, fixed tissue was embedded in Tissue-Tek O.C.T. Compound (#4583, Sakura) and sectioned at a thickness of 10 mm. Then, the sections were washed, permeabilized, blocked and incubated with commercial primary antibodies (1:50, Mouse anti-Sox2 antibody, sc365823, Santa Cruz Biotechnology; 1:200, Rabbit Anti-MASH1/Achaete-scute homolog 1 (ASCL1) antibody, ab211327; 1:200, Rabbit Anti-KI67 antibody, ab15580, Abcam; 1:75, Goat Anti-POMC antibody, ab32893, Abcam; 1:50, Mouse Anti-AR antibody, sc-7305, Santa Cruz Biotechnology; Mouse Anti-GH antibody, sc-374266, Santa Cruz Biotechnology; 1:200, Rabbit Anti-NKX2-2 antibody, ab191077, Abcam; 1:500, Rabbit Anti-Collagen III antibody, ab7778, Abcam; 1:50, Mouse Anti-PIT1 antibody, sc-25258, Santa Cruz Biotechnology; 1:200, Rabbit Anti-TSHβ antibody, ab155958, Abcam; 1:50, Mouse Anti-PRL antibody, sc-46698, Santa Cruz Biotechnology; 1:100, Rabbit Anti-FSHβ antibody, ab180489, Abcam; 1:100, Rabbit Anti-LHβ antibody, ab150416, Abcam) at 4 °C overnight. We used commercial secondary antibodies that were incubated for 2 h at room temperature. Finally, the sections were counterstained with DAPI in an antifade solution (P36931, Invitrogen) and then mounted. The samples were imaged by using an A1RSi+ Nikon confocal microscope (Nikon, Japan).

**Statistics and reproducibility**. For each representative immunohistochemistry and immunofluorescence assay, we took the nearby 1–2 weeks as biological replicates ($n \geq 2$) due to sampling limitations. The positive cells in different sections ($n \geq 3$) were counted automatically by ImageJ software or manually.

**scRNA-seq library construction and sequencing**. We used a mouth pipette to randomly pick single cells and used a modified STRT-seq protocol to construct a scRNA-seq library[25,60]. The cells were lysed in a lysis buffer with an 8-nt cell barcode and 8-nt UMI. Then, mRNA was reverse transcribed into cDNA with SuperScript™ II Reverse Transcriptase (18064014, Invitrogen). After preamplification, the samples with different cell barcodes were pooled together, and the mixture was labeled with a biotin modification by 4 cycles of PCR. Then, the full-length cDNA was sheared into fragments with approximately 300-bp lengths by Covaris (S2), and the 3' cDNA was enriched by Dynabeads MyOne Streptavidin C1 (65002, Invitrogen) to construct a library according to the Kapa Hyper Prep Kit protocol (KK8505, Kapa Biosystems). Cleaned libraries were sequenced as paired-end 150-base reads on an Illumina Hiseq platform (sequenced by Novogene). As individual fetal samples were collected at different time points, they were subjected to different sequencing runs.

**scRNA-seq data processing**. Cells were split by the first 8-bp barcode in Reads2, and then the next 8-bp UMIs in Reads2 were added to the header in Reads1. After trimming the template switch oligo (TSO) and polyA sequences and removing the short reads (length < 37 bp) and low-quality reads ($N > 10\%$) in Reads1, the clean reads were aligned by Tophat (version 2.0.12) to the hg19 genome downloaded from UCSC. Then, HTSeq was applied to count the uniquely mapped reads[61], and the number of different UMIs for each gene in each cell was considered the transcriptional count.

To filter out low-quality cells and multiple cells sequenced as one cell, we selected only cells with a gene number ≥ 2000, an initial reads number ≤ 1E7, and a mapping ratio ≥ 20% and genes with at least one count in at least three cells for the following analysis. The filtered gene expression data set of transcriptional counts was analyzed using the Seurat (Version 2.3.4) package[12]. As most of the single cells in our data sets were around 100,000 transcriptional counts, the scale factor was set to 100,000. The resolution of "FindClusters" was set as 1 for all cells and merged subclusters of MC with the same known markers (Supplementary Fig. 1d, f), 1.5 for endocrine cells and merged subclusters of Stem, Somatotrope and Gonadotrope with the same known markers (Fig. 1b, c), 0.3 for stem cells (Fig. 2c), 0.4 for gonadotropes (Fig. 7a). Main cell types were identified by the combination of known markers for each cluster.

**Identification of DEGs and enrichment analysis**. DEGs were identified by the function "FindAllMarkers" or "FindMarkers" in Seurat packages using "wilcox" test methods and Bonferroni correction. Significant DEGs were selected from genes with adjusted $P$ value p_val_adj ≤ 0.01 and log processed average fold change avg_logFC ≥ 0.5 for further analysis and visualization. In venn diagram of Fig. 6d, TFs were selected from significantly upregulated DEGs (p_val_adj ≤ 0.01 and

avg_logFC ≥ 0.5) between each two of lineages. TFs of each lineage were the union of upregulated genes ($P \leq 1E-4$) compared to the other two lineages, and then the intersection of upregulated genes ($P \leq 1E-4$) compared to the other two lineages were selected as the specific TFs of that lineage. GO analysis and KEGG pathway enrichment analysis of these significant DEGs were performed by Metascape (http://metascape.org)[62]. Pathway enrichment comparisons of each combination of two clusters were analyzed by GSEA[63,64].

**Remove cell cycle effect**. To remove cell cycle effect in the non-proliferative stem cell, we firstly ran a PCA on cell cycle genes in Seurat package (s.genes, g2/m.genes) of stem cells and observed a little cell cycle effect in some stem cells. Then, we regressed out cell cycle scores (S.Score and G2/M.Score), and re-ran a PCA on cell cycle genes and found no cells were separated by these genes. Next, we found clusters using the newly scaled data after regressing out cell cycle scores, and revealed that the newly identified clusters were same as the originally found stem subtypes.

**Gene score definitions**. We defined the E.score, the M.score and the S.score by averaging the expression levels of curated epithelial markers, mesenchymal markers collected from previous studies, and stemness genes in the GO term "stem cell population maintenance" (GO: 0019827) respectively[25,26]. Markers of these scores were listed in Supplementary Data 2.

**Prediction of activated TFs**. The modules of TFs were identified by the SCENIC[17] python workflow (version 0.9.1) using default parameters (http://scenic.aertslab.org). A human TF gene list was collected from the resources of pySCENIC (https://github.com/aertslab/pySCENIC/tree/master/resources), animal TFDB[65,66] (http://bioinfo.life.hust.edu.cn/HumanTFDB#!/download) and the Human Transcription Factors[67] database (http://humantfs.ccbr.utoronto.ca/download.php). Activated TFs were identified in the AUC matrix, and differentially activated TFs were selected using "FindAllMarkers" of the Seurat package. The top 10 enriched activated TFs were ranked by -log10($p\_value$) and demonstrated using the binary matrix (1 activated; 0 not activated). Networks of the modules with TFs and their target genes were visualized by the R package igraph (version 1.2.5) (https://igraph.org/).

**Construction of lineage trajectories**. Lineage trajectories were constructed by Slingshot[29] with a UMAP or PCA plot as the dimensionality reduction results. For Fig. 4a, the start cluster was set as Stem1, and end clusters were mature hormone producing cell types. For Fig. 7a, the start cluster was set as Pre.Gonado. The trajectories were considered the developmental pseudotime of each lineage. TF dynamics along the pseudotime axis were identified by the R package gam (version 1.16)[68], and significantly changed TFs were selected from TFs with $P$-values ≤ 1E-5. In Fig. 4d, upregulated and downregulated TFs were two clusters of candidate TFs identified by hierarchical clustering of genes in Fig. 4b.

**RNA velocity analysis**. The directions of pseudotime were predicted by RNA velocity[28] using exonic and intronic gene expression levels. After alignment by Tophat, mapped bam files were processed by the python package velocyto (version 0.9.1) to produce loom files with spliced and unspliced gene counts. Then, the loom files were merged and analyzed to predict directions following the analysis pipeline with a k-nearest neighbor ($k = 10$). The directions of RNA velocity were projected in a UMAP plot.

**Cell–cell interaction analysis**. Interactions between pairwise cell clusters were inferred by CellPhoneDB v.2.0[69], which includes a public repository of curated ligands, receptors and their interactions. We ran the CellPhoneDB framework using a statistical method and detected L-R pairs that were expressed in more than 20% of cells. Significant L-R pairs ($P$-value ≤ 0.05 and mean value ≥ 0.5) were demonstrated using igraph and heatmap. Cell types expressing ligands were considered as active cell types sending signals, while cell types expressing the corresponding receptors were considered as target cell types receiving signals.

**Integrating human and rodent datasets**. Mouse and rat data sets were downloaded from GEO website GSE120410 and GSE132224, respectively. Cell types were identified by Seurat in each data set independently, and then DEGs were identified by the function "FindAllMarkers" of Seurat packages using "wilcox" test methods and Bonferroni correction. Significant DEGs of human data were selected from genes with p_val_adj ≤ 0.01 and avg_logFC ≥ 0.5. For identifying the potential species-specific genes, we firstly identified cell-type-specific genes (the corticotropes, somatotropes, lactotropes, thyrotropes and gonadotropes) for each species and then made comparision between human and rodent. The RESCUE[70] (version 1.0.1) method was applied for imputing the dropouts for the rodent data. The rodent genes required to be both the mouse and the rat genes. As gene number and gene expression level were much lower in rodent data sets, significant DEGs of rodent data sets were selected from genes with p_val_adj ≤ 0.01 and avg_logFC ≥ 0.25.

To compare the three data sets further, we integrated them by Seurat3 (Version 3.1.1) standard workflow. We only used genes shared in all three data sets. Top 2000 variable features were selected by variance stabilizing transformation ("vst"), and then anchors were identified and passed to the "IntegrateData" function to integrate them, which also removed batch effect among these data sets. After scaling the integrated data and running PCA and UMAP, we clearly observed the relationships among them in Fig. 8a. We also used this workflow to remove batch effect in somatotropes and lactotropes from late stages with more than 10 cells.

**Reporting summary**. Further information on research design is available in the Nature Research Reporting Summary linked to this article.

## Data availability
The authors declare that all data supporting the findings of this study are available within the article and its supplementary information files or from the corresponding author upon reasonable request.

The raw data have been deposited in the GSA (Genome Sequence Archive) databases of the National Genomics Data Center (NGDC, https://bigd.big.ac.cn/) under the BioProject accession code: PRJCA003249. The gene expression matrix data have been deposited in both the GSA and the Gene Expression Omnibus (GEO) under accession code: GSE142653. There are no restrictions on access to these data.

Gene expression patterns of the endocrine cells are also available on the shiny webpage: https://tanglab.shinyapps.io/Human_Fetal_Pituitary_Endocrine_Cells/.

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

## Acknowledgements
The authors thank Dr Hongbo Yang at Department of Endocrinology, Peking Union Medical College Hospital for insightful discussion. We thank Chunyan Shan from the Core Facilities at the School of Life Sciences of Peking University for assistance with confocal microscopy, as well as High Performance Computing Platform of the Center for Life Science at Peking University for assistance with computing analysis. This work is supported by the National Key R&D Program of China (2018YFC1003101, 2018YFA0107601 and 2017YFA0103402), the National Natural Science Foundation of China (31871457, 81521002), the Research Units of Comprehensive Diagnosis and Treatment of Oocyte Maturation Arrest (2018RU001, Chinese Academy of Medical Sciences), the Beijing Municipal Science and Technology Commission (Z181100001318001), and Beijing Advanced Innovation Center for Genomics (ICG) at Peking University.

## Author contributions
L.W., J.Q., T.F. and S.Z. conceived the project. L.W. and S.Z. wrote the manuscript with help from all of the authors. Y.C., J.Y., M.Y. and J.R. performed scRNA-seq. Y.C. and X.M. performed immunofluorescence, immunohistochemistry and imaging. S.Z. conducted the bioinformatics analyses. All of the authors edited and proofread the manuscript.

## Competing interests
The authors declare no competing interests.
