## [Peer Review File · Nature Communications]

Reviewers' Comments:

Reviewer #1:

Remarks to the Author:

In this study, the authors present for the first time scRNAseq of over 4,000 individual cells from whole male and female human fetal pituitaries between ages 7-25 weeks postfertilization. Previous work on this topic was done in rat and mouse pituitaries, and this is acknowledged in the manuscript, as well as in the mouse posterior pituitary (eNeuro 2020; 10.1523/ENEURO.0345-19.2019). The authors characterized developmental trajectories with distinct transitional intermediate states in hormone producing cell lineages and stem/progenitor cells. They confirmed that corticotropes are the first lineage to developed and that the PIT1 lineage (somatotropes, lactotropes and thyrotropes) segregate from a common progenitor coexpressing lineage-specific transcription factors. They also presented novel findings indicating that gonadotropes exhibit a multistep developmental trajectory, including a novel fetal gonadotrope cell subtype expressing chorionic gonadotropin gene. The major findings they presented in this manuscript relates to characterization of the cellular heterogeneity of pituitary stem cells and identification of a hybrid epithelial/mesenchymal state and an early-to-late state transition. Several scRNAseq results were also validated using immunocytochemistry. Finally, they provide a nice comparison of human, rat, and mouse scRNAseq data.

Major concerns:

The authors list nonpituitary cell types and pituitary endocrine cells, but do not discuss or present any data on expression of pituitary glia-like folliculostellate cells (FSCs), which are also present in the postnatal human pituitary. It is possible that these cells are not developed during 25 weeks of embryonic life. If that is the case, the authors should provide evidence and state it.

The authors define stem cells based on expression of two genes, SOX2 and PROP1. However, in postpubertal rat pituitaries, these genes are found in FSCs, which were defined as cells expressing S100b, Fxyd1, Gstm2, and Capn6. Are those genes expressed in human stem cells? If a fraction of "stem" cells express FSC marker genes, they should be shown separately.

A cluster of 90 cells was identified as posterior pituitary cells, based on expression of OTX2 gene. Posterior pituitary is a complex tissue, composed of several cell types, including pituicytes. From presented data, it is not clear whether development of posterior pituitary has not been completed during 25 weeks of embryonic life, the tissue was not collected, or the cell dispersion procedure was not appropriate for pituicytes.

The definition of corticotropes using only POMC and TBX19 does not distinguish melanotropes from corticotropes. The authors should determine whether melanotropes were present or not in embryonic pituitary tissue, and the result should be stated in L. 253-279.

If cell type cluster identities are changed due to any of the above considerations, then all results that depend on these clusterings (differential gene expression, relationships between cell types, etc.) could possibly change. This could alter some of the results and conclusions of the manuscript.

While Fig S1b and c lend a small amount of confidence toward similarity of samples across batches, it is not clear how Figs S1d and S2a demonstrate that there were minimal batch effects between samples (Line 85). How many different Illumina sequencing runs were performed? Were the 21 samples randomized and distributed among those runs? This important part of the experimental protocol should be clearly stated.

The authors' application of the Slingshot methodology for lineage definition does not appear to be consistent with RNA velocity nor with the known age of samples. If the direction of trajectories is inferred from RNA velocity, then the stem cell trajectory is in the wrong direction in Fig 4a, right

panel. It seems from several results shown in the paper, including age of samples and RNA velocity, that the origin of lineages should be the early Stem cluster, Stem1, not Stem3 as the authors have presented. Conclusions that depend on these lineages are therefore questionable. This can be addressed during lineage computation because Slingshot supports user-specified clusters as the beginning and ends of trajectories. The terminal clusters are also known in this case, based on the mature expression of known cell-type markers. Adding trajectory endpoints, when known, promotes a more robust and biologically meaningful result, as recommended by the authors of Slingshot (Kelly et al. 2018, PMC6007078).

Regarding the corticotrope lineage, Fig S2a clearly shows that at the earliest time-point, the corticotrope cluster is already separate from the rest of cells. The cells between those corticotropes and the nearest other cell cluster (pre-gonadotropes) appeared later, casting doubt on the computational lineage connection from Pro-PIT1/Stem to corticotropes in this dataset. This connection may indeed be present but observed at a time point earlier than 7wks.

Fig S2a seems to be a very important and informative panel and should probably appear in a main figure (eg. Fig 1 or Fig 4a).

Data availability – the link provided is currently broken. It would be a tremendous asset to have the scRNAseq data of this study submitted to the Genome Expression Omnibus (GEO). Will this be done?

Other comments:

The computational data processing methods are incompletely described. Please clearly list in methods the sequence of Seurat commands used to obtain results, including normalization, highly variable feature selection, PCA, clustering, tSNE and UMAP. Each of these methods has adjustable parameters, so please include the parameter values used, or state that defaults were used if that was the case. If parameters differed for specific results/figures, state the value used in each case instead of a range (e.g. line 626: FindClusters [resolution] between 0.1 and 2.0 - state what values were used for each of the different clusterings)

Were any results in the paper other than Fig S3 dependent on regressing out cell-cycle genes? If so, please clearly state when this was done and why. For example, Fig 4a, right panel, is missing the CC cluster. Is this because in that case CC effects were regressed out?

In many places the authors refer to "TFs" when it appears that the more general "genes" was intended. For example - it seems unlikely that differential gene expression analysis between cell types was performed only using TF genes. For example, in Fig 1e TFs are highlighted, but are all 10 markers per cell type really TFs? Similarly, in Fig 6c - PRL is a hormone, not a TF.

L. 85: Reference to incorrect panel Fig S1b. Perhaps the authors intended S1c?

L. 102: missing reference for PDGFRA as a marker gene for MC cells.

L. 116: Fig. S1h is missing.

L. 142: It is not clear how Fig S3d,e show that clustering remained similar after regressing out cell-cycle genes. In fact, these panels don't show any clear distinction between Stem1-3 in the PC1-PC2 plane.

L. 127: A reference is needed to support the claim that listed genes are marker genes for stem cells. In the rat model, Zfp3611, Anxa1, and Nfib are expressed in FSCs, and Nr2f2 is expressed in all pituitary cell types.

L. 157: "the bHLH TF ASCL1" term is confusing. Maybe Basic helix-loop-helix protein TF gene Ascl1.

L. 194: Could you provide (perhaps as supplementary table) the genes used to define the E-, M-, and S-scores?

L. 186-190: Reference for these marker genes should be added.

L. 285-288: Any information about Pre.Lac?

L. 349-355: The lack of lactomatotrophs was also observed in the rat model, and this should be acknowledged.

L. 483: GATA2+ comparison with thyrotropes is missing.

L. 731: remove the first "are" – cells mainly belong...

Reviewer #2:

Remarks to the Author:

This manuscript presents the first comprehensive analysis of human fetal pituitary gland gene expression using single cell RNA sequencing. The authors utilized 21 fetuses staged 9-27 wk gestation, which represents a time in which pituitary stem cells have already become engaged towards differentiation and express key transcription factors such as SOX2 and PROP1. During this time period progenitors are actively committing to all of the pituitary hormone producing cell fates. This permits the authors to capture cells that are in the process of differentiation. As such, this is a valuable, rich data resource for generating hypotheses about pituitary development. The authors present several hypotheses, but many of them are quite speculative, and it is difficult to be certain whether there is enough depth in the dataset to support these assertions.

The number of cells sequenced at each time point may be insufficient to support some of the authors conclusions. For example, it appears that no corticotropes were captured at 16 wk or 21 wk. While there are clearly trends, such as reduced numbers of Pro-PIT1 cells and proliferating stem cells later stages, the proportions of various cell types vary considerably even amongst similar gestation times (compare 15, 16 and 17 wk, Fig. 1). Some conclusions need to be tempered with this in mind.

The authors are to be congratulated for validating some of their findings in single cell RNA seq with immunohistochemical staining for proteins (Figs 2, 5, 6). The authors argue that stem cells and mesenchymal cells are interacting based on gene ontology terms for ligands and receptors. It would be more convincing if they could show a table with the actual ligand and receptor pairs in these cells, rather than relying on GO terms. They do show immunostaining that co-localizes a mesenchymal and a stem cell marker, but data on the actual ligand/receptor pairs could provide important support.

Fig. 3 presents an analysis of epithelial to mesenchymal-like transition (see also Supplementary Fig. 4). Studies in mice show that SOX2 is initially expressed throughout the pituitary primordium or Rathke's pouch. Later, PROP1 is activated in many SOX2 expressing cells, and PROP1, either directly or indirectly, suppresses SOX2 expression, which drives a very small portion of cells lining the marginal zone, or stem cell niche, to delaminate and migrate to form the anterior lobe. The authors argue strongly that the human pituitary stem cells exist in a hybrid epithelial-mesenchymal state. It seems plausible that they have not looked early enough to capture the SOX2+, PROP1- early progenitors and that the shallow sequencing has missed the subpopulation of cells that are transitioning. This alternate interpretation should be considered. Predicted

transcription factor binding to EMT markers seems too speculative. How valid is it to use scenic to identify transcription factors and their regulons? There is a paper by Holland et al in Genome Biology 2020 that implies other programs may be more robust than scenic. Regardless of the software used, this approach too speculative.

Fig. 4 presents genes whose expression changes over time for five hormone-producing cell types. The stem cell markers do exhibit a sharp drop in expression (Panel C), and some of the lineage specific transcription factors show a sharp increase (POU1F1, TBX19, NR5A1), but there seems to be a large amount of scatter for many of the factors presented. Perhaps this figure should be simplified to highlight only the factors that are readily interpretable. The description in the results seems to exaggerate the clarity.

Fig. 5 presents the trajectory of corticotrope differentiation and identification of androgen receptor as a marker. This is convincing. Fig. 6 presents the differentiation of the PIT1 lineages. This seems much less convincing. Liebhaber's analysis of mouse somatotrope and lactotrope expression, PMC6260062, showed that these cells have very similar transcriptomes in mice. This is not unexpected because of the evolutionary conservation of these hormones. The authors find that lactotropes and thyrotropes have the most similar transcriptomes. This is unexpected and inconsistent with the mouse results. Please discuss. Along the same lines, the authors refer to ZBTB20 as ubiquitous and lactotrope specific. From the data in Fig. 6C it does not seem to be enriched.

Fig. 7 presents the stepwise differentiation of gonadotropes. This is consistent with reports in mice. However, the diagram in Fig. 7e is confusing because it highlights transcription factors that may or may not be enriched for the stages shown.

Fig. 8. The mechanism for comparing adult rodent gene expression data sets with the fetal human dataset seems problematic. It is also not clear why the number of human specific genes should be greater than rodent. Please discuss false negatives. What fraction of the reads did not map back to the genome?

Supplemental Fig. 2 nicely illustrates the onset of expression of pituitary hormones during gestation (2b). Panel 2c shows the enrichment of the top 10 transcription factors for each cell type. Other than the stem cells and corticotropes, the enrichment for other cell types seems very noisy. This is also true for the comparison of the PIT1 lineages in Supplementary Figure 5. These figures contrast with Supplementary Fig. 6 where the gene expression profiling of the stages of gonadotropes seems more clear.

Technical considerations:

- Is 4000 cells enough? Does it make sense that there are 4506 genes expressed and ~86K transcripts?
- Does the correction for batch effects make sense? In the supplement they show similar GAPDH expression across sets.
- How was the stem cell reclustering done? Was expression of SOX2 the basis for calling these stem cells?

Minor things:

- L. 166 refers to WNT5A as canonical WNT signaling, but it usually acts in a non-canonical fashion.
- Line 187 cadeharin – cadherin
- l. 248 satges – stages
- l. 325 neruoD4 – neuroD4
- Present Zbtb20 expression as ubiquitous yet lactotrope specific. Please clarify.

- I. 309 Gata2 ko references Dasen et al., which described the effects of a dominant negative Gata2 transgene and ectopic Gata2 expression. This is a valid reference. The authors ref. 49 is the Gata2 ko.
- In the discussion of mammo-somatotropes Nasonkin et al., is cited as a reference. This paper does not present evidence for cells expressing both prolactin and growth hormone.
- There are numerous errors in the listing of authors in the references.
- Supplemental Fig. 1d. Is there a better way to illustrate males and females? The circles and triangles are difficult to discern.
- Supplemental Fig. 1 g. presents 8 disease terms for 19 genes. It would be more accurate to describe the consequences of mutations in these genes than to use these overlapping, sketchy terms.

We thank the reviewers for their thorough and critical examination of our manuscript. We also appreciate their helpful suggestions and comments, which have been incorporated into our revised manuscript. Responses to specific comments are listed below.

Responses to the comments of Reviewer 1:

In this study, the authors present for the first time scRNAseq of over 4,000 individual cells from whole male and female human fetal pituitaries between ages 7-25 weeks postfertilization. Previous work on this topic was done in rat and mouse pituitaries, and this is acknowledged in the manuscript, as well as in the mouse posterior pituitary (eNeuro 2020; 10.1523/ENEURO.0345-19.2019). The authors characterized developmental trajectories with distinct transitional intermediate states in hormone producing cell lineages and stem/progenitor cells. They confirmed that corticotropes are the first lineage to develop and that the PIT1 lineage (somatotropes, lactotropes and thyrotropes) segregate from a common progenitor coexpressing lineage-specific transcription factors. They also presented novel findings indicating that gonadotropes exhibit a multistep developmental trajectory, including a novel fetal gonadotrope cell subtype expressing chorionic gonadotropin gene. The major findings they presented in this manuscript relates to characterization of the cellular heterogeneity of pituitary stem cells and identification of a hybrid epithelial/mesenchymal state and an early-to-late state transition. Several scRNAseq results were also validated using immunocytochemistry. Finally, they provide a nice comparison of human, rat, and mouse scRNAseq data.

Response:

We thank the reviewer for these positive comments.

Major concerns:

1. The authors list nonpituitary cell types and pituitary endocrine cells, but do not discuss or present any data on expression of pituitary glia-like folliculostellate cells (FSCs), which are also present in the postnatal human pituitary. It is possible that these cells are not developed during 25 weeks of embryonic life. If that is the case, the authors should provide evidence and state it. The authors define stem cells based on expression of two genes, *SOX2* and *PROP1*. However, in postpubertal rat pituitaries, these genes are found in FSCs, which were defined as cells expressing *S100b*, *Fxyd1*, *Gstm2*, and *Capn6*. Are those genes expressed in human stem cells? If a fraction of “stem” cells express FSC marker genes, they should be shown separately.

Response:

We are thankful to the reviewer for the suggestions and comments. We identified a total of 14 cell clusters including non-pituitary cell types and pituitary endocrine cells. These 14 cell clusters were clearly defined to certain cell types, thus indicating that the folliculostellate cells (FSCs) do not present as a distinct cell cluster in our data. This is consistent with previous immunostaining studies showing that S100-positive FSCs appear only in the postnatal rat pituitary. According to the reviewer’s suggestion, we examined the expression of the FSC markers *S100B*, *FXYD1*, *GSTM2* and *CAPN6* to investigate any possible presence of the FSCs, particularly as a subpopulation of the stem cells. These four genes were expressed in a variety of cell types in the human fetal pituitary, in a manner similar to the rat and mouse pituitary cells (Figure R1). In the human fetal pituitary stem cells, *S100B*, *FXYD1*, *GSTM2* and *CAPN6* were expressed in 3.4%, 16.8%, 30.8% and 10.3% of the cells, respectively (NEW Fig. 1c and Figure R2 for expression of *S100B*). However, no cells are positive for all four genes. We also examined the differentially expressed genes (DEGs) between the *S100B*-positive and the *S100B*-negative stem cells and only identified two genes, *SH3GL3* and *MYH15*, which are not enriched in the rat FSCs (Figure R3). Together, these data suggested that the FSCs were not present in the human fetal pituitary during 25 weeks post-fertilization. The description has been added to the new manuscript L181-L187.

Figure R1. Violin plots showing gene expression of *S100B*, *FXYD1*, *GSTM2* and *CAPN6* in the human pituitary, which were not enriched in the stem cells.

Figure R2. Scatterplots showing that only a few pituitary stem cells expressed *S100B*.

Figure R3. Violin plots showing that *SH3GL3* and *MYH15* were not enriched in the rat FSCs.

2. A cluster of 90 cells was identified as posterior pituitary cells, based on expression of *OTX2* gene. Posterior pituitary is a complex tissue, composed of several cell types, including pituicytes. From presented data, it is not clear whether development of posterior pituitary has not been completed during 25 weeks of embryonic life, the tissue was not collected, or the cell dispersion procedure was not appropriate for pituicytes.

Response:

According to the markers *OTX2*, *LHX2*, *RAX* and *COL25A1*, the cluster of 90 posterior pituitary cells should be pituicytes (NEW Supplementary Fig. 1f for showing *RAX* expression in addition to *OTX2*). Since we have focused on the anterior pituitary in the present study, we have not systematically collected the posterior pituitary cells, and analyzed the posterior pituitary only in some fetuses. Specifically, for fetuses earlier than 14 weeks, the whole pituitaries, including the anterior pituitaries and potential posterior pituitaries, were analyzed. For fetuses later than 14 weeks, we separated the anterior and posterior pituitaries, and only analyzed the anterior pituitaries except two fetuses (15W1 and 17W1), of which we collected cells from both parts. A total of 58 high-quality cells

were obtained from the posterior pituitaries of these two fetuses, most of which (n = 43) were pituicytes and other cells included 12 immune cells and 3 red blood cells. The data and description have been added to the Methods section, L595-L600.

3. The definition of corticotropes using only POMC and TBX19 does not distinguish melanotropes from corticotropes. The authors should determine whether melanotropes were present or not in embryonic pituitary tissue, and the result should be stated in L. 253-279.

Response:

The melantrope appears after 14 weeks of gestation but is scarce in the human pituitary. The melantrope coexpresses *TBX19* and *PAX7*, both of which are required for its differentiation. Only 4 cells, which were 7- or 8-week stem cells, coexpressed *TBX19* and *PAX7* in our data (NEW Fig. 1c and Figure R4). Comparison between these cells and other stem cells revealed only *PAX7* and three other DEGs (*RASSF6*, *IL23A* and *AMIGO2*) that were not expressed in the mouse and rat melantropes (Figure R5). These results suggested that we have not captured the melantrope population. We have added the description in the Result section, L269-L275.

Figure R4. Scatterplots showing that no *TBX19*-positive corticotropes expressed *PAX7*.

Figure R5. Violin plots showing that *Rassf6*, *Il23a* and *Amigo2* were not expressed in mouse melanotropes.

4. If cell type cluster identities are changed due to any of the above considerations, then all results that depend on these clusterings (differential gene expression, relationships between cell types, etc.) could possibly change. This could alter some of the results and conclusions of the manuscript.

Response:

The above analyses have excluded the presence of FSCs and melanotropes in our dataset. Thus the results and conclusions of the manuscript were not changed.

5. While Fig S1b and c lend a small amount of confidence toward similarity of samples across batches, it is not clear how Figs S1d and S2a demonstrate that there were minimal batch effects between samples (Line 85). How many different Illumina sequencing runs were performed? Were the 21 samples randomized and distributed among those runs? This important part of the experimental protocol should be clearly stated.

Response:

Supplementary Fig. 1d and Supplementary Fig. 2a (NEW Fig. 1b) displayed how different fetal samples were distributed on a given cell cluster. The results showed that each of 14 cell cluster was composed of multiple fetal samples, thus indicating that these cell clusters were not consequence of batch effects. We considered that if any cluster was resulted from batch effects of a given fetal sample, it will be solely composed of that sample. These figures also showed that, within each cell cluster, the samples of the similar stages were largely mixedly distributed. It should be noted that the cells of the early stages may be biologically different from those of the late stages, and thus the different distribution of the early and late stage cells in these figures were supposed to reflect real biological differences, but not batch effects. In Figure R6, we used the histogram to display the results that each cell cluster contains multiple fetal samples (Figure R6); the histogram of Fig. 1d in the manuscript showed a similar result. Batch effects are complex and we agree that Supplementary Fig. 1b, c, d, 2a only addressed the batch effects on certain aspects and thus we cannot claim that these results can demonstrate minimal batch effects in all aspects. Thus we have more directly described

the results as “The samples gave similar gene numbers and similar expression levels of *GAPDH* across batches” in L86, and “Each cell cluster was composed of multiple fetal samples, and the samples of similar stages, or different sexes, were largely mixedly distributed” in L109-L110.

Figure R6 Bar plots showing the proportions of each week in each cell type.

6. The authors' application of the Slingshot methodology for lineage definition does not appear to be consistent with RNA velocity nor with the known age of samples. If the direction of trajectories is inferred from RNA velocity, then the stem cell trajectory is in the wrong direction in Fig 4a, right panel. It seems from several results shown in the paper, including age of samples and RNA velocity, that the origin of lineages should be the early Stem cluster, Stem1, not Stem3 as the authors have presented. Conclusions that depend on these lineages are therefore questionable. This can be addressed during lineage computation because Slingshot supports user-specified clusters as the beginning and ends of trajectories. The terminal clusters are also known in this case, based on the mature expression of known cell-type markers. Adding trajectory endpoints, when known, promotes a more robust and biologically meaningful result, as recommended by the authors of Slingshot (Kelly et al. 2018, PMC6007078).

Response:

We thank the reviewer for this helpful suggestion. In fact, the velocity analysis suggested that the start point are the stem cells of the early stage. We have changed the start point of the trajectories as Stem1 instead of all stem cells. We have reanalyzed all the data based on this change (relating to NEW Fig. 4a, b, c and d, NEW Fig. 6a, b and f; see also Figure R7 and R8).

Figure R7. Pseudotime analysis of all endocrine cells shown in the UMAP plot. Yellow circle in the right panel represents the start point of the trajectories which was set as the Stem1 subcluster. Note that the stem cells directly connected to the corticotropes and did not cross the pre-gonadotropes.

Figure R8. Heatmap showing the relative expression of TFs displaying significant changes ($P \leq 1E-5$) along the pseudotime axis of each lineage. Note that the start point has been changed to the early stem cells.

7. Regarding the corticotrope lineage, Fig S2a clearly shows that at the earliest time-point, the corticotrope cluster is already separate from the rest of cells. The cells between those corticotropes and the nearest other cell cluster (pre-gonadotropes) appeared later, casting doubt on the computational lineage connection from Pro-PIT1/Stem to corticotropes in this dataset. This connection may indeed be present but observed at a time point earlier than 7wks.

Response:

We thank the reviewer for this helpful suggestion. After changing the start point of the trajectory to the early stem cells, this mistake was remarkably corrected. In the results, the stem cells directly connected to the corticotropes and did not cross the pre-gonadotropes (NEW Fig. 4a and Figure R7).

8. Fig S2a seems to be a very important and informative panel and should probably appear in a main figure (eg. Fig 1 or Fig 4a).

Response:

We have moved Supplementary Fig. 2a to NEW Fig. 1b.

9. *Data availability – the link provided is currently broken. It would be a tremendous asset to have the scRNAseq data of this study submitted to the Genome Expression Omnibus (GEO). Will this be done?*

Response:

We have actually already uploaded the data on the GEO webpage in the initial submission, which is under the accession number GSE142653 (<https://www.ncbi.nlm.nih.gov/geo/query/acc.cgi?acc=GSE142653>, the secure token is uncjmaacvrkrlon). We greatly apologize for missing it in the OLD manuscript. We have added this important information in the Methods section (Data availability) L750-L752.

Other comments:

10. *The computational data processing methods are incompletely described. Please clearly list in methods the sequence of Seurat commands used to obtain results, including normalization, highly variable feature selection, PCA, clustering, tSNE and UMAP. Each of these methods has adjustable parameters, so please include the parameter values used, or state that defaults were used if that was the case. If parameters differed for specific results/figures, state the value used in each case instead of a range (e.g. line 626: FindClusters [resolution] between 0.1 and 2.0 - state what values were used for each of the different clusterings)*

Response:

We have added description of these parameters in the Method section, L661-664.

11. Were any results in the paper other than Fig S3 dependent on regressing out cell-cycle genes? If so, please clearly state when this was done and why. For example, Fig 4a, right panel, is missing the CC cluster. Is this because in that case CC effects were regressed out?

Response:

We only regressed out 96 cell-cycle genes for the results of Fig S3d and S3e, which examined the effects of cell cycling on subclustering the stem cells (NEW Supplementary Fig. 3d). In Fig 4a, the CC cluster was directly omitted, but not by regressing out the cell cycling genes. We found that present of the CC cluster caused wrong trajectories in the pseudotime analysis (Figure R9). Also, Fig 4a was directly linked to Fig 4b. Since the CC cluster contained the *POUIF1*-positive and *TBX19*-positive cells in addition to the stem cells, we concerned that these cells may also affect the trajectories of these cell lineages in Slingshot analysis. We have added the description in the legend of Fig. 4, L811-L813.

Figure R9. The pseudotime trajectories of Slingshot analysis including the CC cluster.

12. *In many places the authors refer to "TFs" when it appears that the more general "genes" was intended. For example - it seems unlikely that differential gene expression analysis between cell tyupes was performed only using TF genes. For example, in Fig 1e TFs are highlighted, but are all 10 markers per cell type really TFs? Similarly, in Fig 6c - PRL is a hormone, not a TF.*

Response:

We have carefully checked the manuscript and changed the improperly applied “TFs”, including the text corresponding to Fig. 6c. In Fig. 1e, the SCENIC analysis focused on the TFs, and so all markers per cell type were TFs. The gene list of Fig. 1e (displaying TF activity) was actually exactly corresponding to that of Supplementary Fig. 2c (displaying gene expression), which showed that all these genes were TFs. We apologize that the description in the OLD manuscript was not clear and we have added the description in the legend of Fig. 1e and Supplementary Fig. 2c in the NEW manuscript (L132). The texts relating to Fig. 4 focused on TFs and thus we kept the use of TFs to make it clear.

13. *L. 85: Reference to incorrect panel Fig S1b. Perhaps the authors intended S1c?*

Response:

We have corrected this in the new manuscript (L86).

14. *L. 102: missing reference for PDGFRA as a marker gene for MC cells.*

Response:

We have added the reference in the NEW manuscript (L104).

15. L. 116: Fig. S1h is missing.

Response:

We have corrected it in the NEW manuscript (L120).

16. L. 142: It is not clear how Fig S3d,e show that clustering remained similar after regressing out cell-cycle genes. In fact, these panels don't show any clear distinction between Stem1-3 in the PC1-PC2 plane.

Response:

We apologize for the confusion. The PCA plots were used the cell cycle genes (PC1: 43 S phase genes, PC2: 54 G2M phase genes) to show that the cell cycle genes could not distinguish between the stem cell subclusters. We have used the tSNE plots to show the clustering results before and after regressing out the cell cycle genes in the NEW supplementary Figure 3d (Figure R10). We have add the description in the legend of supplementary Figure 3d.

Figure R10. The tSNE plots (left) showing the subclusters of the stem cells before (upper panels) and after (lower panels) regressing out the cell cycle genes. In addition, the PCA plots (right) using the cell cycle genes (PC1: 43 S phase genes, PC2: 54 G2M phase genes) could not distinguish between these subclusters.

17. L. 127: A reference is needed to support the claim that listed genes are marker genes for stem cells. In the rat model, Zfp3611, Anxa1, and Nfib are expressed in FSCs, and Nr2f2 is expressed in all pituitary cell types.

Response:

As show in the violin plots in Figure R10 (also heatmap in Figure 2A of the manuscript), the expression of *ZFP36L1*, *ANXA1*, *NFIB*, *ZNF521* and *NR2F2* were enriched in the human stem cells. We have added two related references in the NEW manuscript (L133-L134), including the one that were published by Nishimura et al. which reported *ZFP36L1* as a potential regulatory gene of *Prop1* in the mouse stem cells.

Figure R11. Expression of *ZFP36L1*, *ANXA1*, *NFIB*, *ZNF521* and *NR2F2* in the human fetal pituitary.

18. L. 157: “the bHLH TF *ASCL1*” term is confusing. Maybe Basic helix-loop-helix protein TF gene *Ascl1*.

Response:

We have corrected it in the NEW manuscript (L166).

19. L. 194: Could you provide (perhaps as supplementary table) the genes used to define the E-, M-, and S-scores?

Response:

We have provided the gene list in the NEW Supplementary table 2.

20. 186-190: Reference for these marker genes should be added.

Response:

We have added references in the new manuscript (L206).

21. L. 285-288: Any information about Pre.Lac?

Response:

We have not found a cell subcluster of Pre.Lac. Only three putative intermediated progenitor or precursor cell populations were identified including the Pro.PIT1_all cells as a common progenitor for all three hormone producing cell types, the Pre.Thy as a precursor for the thyrotrope and the Pre.Som as a potential precursor for the somatotrope.

22. L. 349-355: The lack of lactomatotrophs was also observed in the rat model, and this should be acknowledged.

Response:

We have added this point in the NEW manuscript (L367-369).

23. L. 483: GATA2+ comparison with thyrotropes is missing.

Response:

Thanks for this suggestion. To distinguish between the thyrotropes and the Pre.Gonado cells, we have revised the description of the cell state as *GATA2⁺/POU1F1/NR5A1/LHB* in the NEW manuscript (L509). We have also added *GATA2* expression in the NEW Figure 1c.

24. L. 731: remove the first “are” – cells mainly belong...

Response:

We have corrected it in the NEW manuscript (L777).

Responses to the comments of Reviewer 2:

This manuscript presents the first comprehensive analysis of human fetal pituitary gland gene expression using single cell RNA sequencing. The authors utilized 21 fetuses staged 9-27 wk gestation, which represents a time in which pituitary stem cells have already become engaged towards differentiation and express key transcription factors such as *SOX2* and *PROP1*. During this time period progenitors are actively committing to all of the pituitary hormone producing cell fates. This permits the authors to capture cells that are in the process of differentiation. As such, this is a valuable, rich data resource for generating hypotheses about pituitary development. The authors present several hypotheses, but many of them are quite speculative, and it is difficult to be certain whether there is enough depth in the dataset to support these assertions.

Response:

We thank the reviewer for the positive comments.

1. The number of cells sequenced at each time point may be insufficient to support some of the authors conclusions. For example, it appears that no corticotropes were captured at 16 wk or 21 wk. While there are clearly trends, such as reduced numbers of Pro-PIT1 cells and proliferating stem cells later stages, the proportions of various cell types vary considerably even amongst similar gestation times (compare 15, 16 and 17 wk, Fig. 1). Some conclusions need to be tempered with this in mind.

Response:

We are thankful to the reviewer for the suggestions and comments. We agree that the number of cells sequenced at some time points was not sufficient. This was due to unpredictable availability and quality of the human fetal samples. We were not able to obtain fetal samples of 11, 12, 18 and 20 weeks. Also, when the analyzed cell number for the 15 and 16 and 21-week fetal samples were not sufficient, we were not able to obtain another fetal sample of the same time point. However, since the numbers of sequenced cells in most time points were satisfactory, we consider that the majority of our conclusions, particularly the identification of cell types, will not be affected. The reason that no corticotropes were captured at 16 weeks or 21 weeks and the proportions of various cell types vary considerably amongst 15, 16 and 17 weeks was due to the low sequenced cells of the 15, 16 and 21 weeks (sequenced endocrine cells number: 32, 17 and 30 for 15, 16 and 21 weeks, respectively, see NEW Supplementary Fig S1a). To avoid these misleading results, we have removed the results of 15-week, 16-week and 21-week fetal samples in Fig. 1d (see NEW Fig. 1d and Figure R1), of which the numbers of the sequenced endocrine cells were lower than 50. Except these three fetal samples, other samples showed acceptable variation of different cell types between neighboring embryonic times, suggesting that the sequenced cell numbers were satisfactory. One conclusion that may be affected is the timing when a hormone-producing cell type first appears. For this, we have used immunostaining to verify the scRNA-seq results. Particularly, PRL immunostaining of a 15-week fetus failed to detect PRL-positive cells, supporting the conclusion that PRL-positive cells first appear at 16 weeks.

Figure R1. Bar plots showing the proportions of each cell type in each stage. Solid circles at the bottom indicate the earliest stages when a hormone producing cell type appears.

2. The authors are to be congratulated for validating some of their findings in single cell RNA seq with immunohistochemical staining for proteins (Figs 2, 5, 6). The authors argue that stem cells and mesenchymal cells are interacting based on gene ontology terms for ligands and receptors. It would be more convincing if they could show a table with the actual ligand and receptor pairs in these cells, rather than relying on GO terms. They do show immunostaining that co-localizes a mesenchymal and a stem cell marker, but data on the actual ligand/receptor pairs could provide important support.

Response:

We thank the reviewer for the suggestion. We have added the gene list of the ligands and receptors pairs in NEW Supplementary Table 1. We agree that immunostaining experiments can provide more supports on the interaction between stem cells and mesenchymal cells. However, due to the COVID-19 pandemic, we have not been yet allowed to come back to Peking University, and thus are not convenient to do such an experiment at present. The interaction between stem cells and mesenchymal cells is a very new research area and we just aim to provide some clues from the scRNA-seq data,

and experiment validation is a bit beyond the scope of the present study. We have toned down the description of this part in L137-L142.

3. Fig. 3 presents an analysis of epithelial to mesenchymal-like transition (see also Supplementary Fig. 4). Studies in mice show that SOX2 is initially expressed throughout the pituitary primordium or Rathke's pouch. Later, PROP1 is activated in many SOX2 expressing cells, and PROP1, either directly or indirectly, suppresses SOX2 expression, which drives a very small portion of cells lining the marginal zone, or stem cell niche, to delaminate and migrate to form the anterior lobe. The authors argue strongly that the human pituitary stem cells exist in a hybrid epithelial-mesenchymal state. It seems plausible that they have not looked early enough to capture the SOX2⁺, PROP1⁻ early progenitors and that the shallow sequencing has missed the subpopulation of cells that are transitioning. This alternate interpretation should be considered.

Response:

We agree with the reviewer that we may have not captured all E/M substates of the pituitary stem cells including the early SOX2⁺/PROP1⁻ cells and other transitioning cells. We have added this points in the Discussion section (L557-L560), and have toned down our claim of this part.

4. Predicted transcription factor binding to EMT markers seems too speculative. How valid is it to use scenic to identify transcription factors and their regulons? There is a paper by Holland et al in Genome Biology 2020 that implies other programs may be more robust than scenic. Regardless of the software used, this approach too speculative.

Response:

The SCENIC method combines the cis-regulatory motif and co-expression module analysis, thus can predict activated TFs with their putative regulatory subnetworks. In Fig. 1e, known TFs of each cell type were active in corresponding cell types, which mean that SCENIC precisely predict the TF modules in our data. We have applied the method

of Holland et al in Genome Biology 2020. However, the method did not reveal specific activation of the known lineage-specific TFs such as *POU1F1*, *TBX19*, *NR5A1* and *GATA2* in the corresponding lineages, suggesting that it was not suitable for the present pituitary data (Figure R2). We agree that these methods are anyway speculative. We have toned down this part in L222-L226. We have also added the description of the SCENIC method at L121-L122.

Figure R2. Heatmap of known TFs activities predicted by DoRothEA. A-E are the confidence scores of five categories in DoRothEA, ranking from A (highest confidence) to E (lowest confidence)

5. Fig. 4 presents genes whose expression changes over time for five hormone-producing cell types. The stem cell markers do exhibit a sharp drop in expression (Panel C), and some of the lineage specific transcription factors show a sharp increase (*POU1F1*, *TBX19*, *NR5A1*), but there seems to be a large amount of scatter for many of the factors presented. Perhaps this figure should be simplified to highlight only the factors that are readily interpretable. The description in the results seems to exaggerate the clarity.

Response:

According to the reviewer's suggestion, we have simplified this figure and only kept the genes that were readily interpretable (NEW Fig. 4b and Figure R3). We have also simplified the corresponding description in the text L241-L247.

Figure R3. Scatterplots showing the expression levels of representative known and novel TFs along the pseudotime axis.

6. Fig. 5 presents the trajectory of corticotrope differentiation and identification of androgen receptor as a marker. This is convincing.

Response:

We thank the reviewer's comments.

7. Fig. 6 presents the differentiation of the PIT1 lineages. This seems much less convincing. Liebhaber's analysis of mouse somatotrope and lactotrope expression, PMC6260062, showed that these cells have very similar transcriptomes in mice. This is not unexpected because of the evolutionary conservation of these hormones. The authors find that lactotropes and thyrotropes have the most similar transcriptomes. This is unexpected and inconsistent with the mouse results. Please discuss.

Response:

To further confirm this, we performed correlation analysis for the transcriptome of all human fetal pituitary endocrine cell types, as well as the mouse and rat adult pituitary endocrine cell types for comparison. The results verified that the thyrotropes and the lactotropes were close to each other relative to the somatotropes. In contrast, the

somatotropes and lactotropes were more close to each other comparing with the thyrotropes in the mouse and rat adult pituitaries, which was consistent with the evolutionary conservation of GH and PRL (NEW Supplementary Fig. 5c, Figure R4). We speculated that the closer relationship is may be due to that the somatotrope develop into a more mature state comparing with the lactotroph and thyrotrophes during 25-week. It is known that fetal GH begin to secrete at 10-week gestation and peaks at 22-week gestation. The PRL begin secrete at 25-week gestation and peaks at birth. Supporting this, we found that *TRHR* and *ESR1* were lowly expressed in the fetal thyrotropes and lactotropes respectively, suggesting that both lineages were not fully matured (NEW Fig. 8c and Figure R5). We have added the description in L353-363.

Figure R4 Heatmap of Pearson correlation among endocrine cell types in fetal human pituitary, adult mouse and rat pituitaries.

Figure R5. Expression of *TRHR* and *ESR1* in the human fetal pituitary and the rat and mouse pituitaries.

8. *Along the same lines, the authors refer to ZBTB20 as ubiquitous and lactotrope specific. From the data in Fig. 6C it does not seem to be enriched.*

Response:

ZBTB20 was expressed in all pituitary cell types. For its expression dynamics along the lactotrope development, its expression level was significantly upregulated in the Pro.PIT1_all in comparison with the stem cell ($\logFC = 0.6$, $P = 9.8E-08$), and the level was similar between the Pro.PIT1_all and the lactotrope ($P > 0.1$, Fig 6c). Due to downregulation in other two lineages, the expression level of *ZBTB20* was slightly but significantly higher in the lactotrope comparing with the somatotrope and the thyrotrope (Lactotrope versus Somatotrope: $\logFC = 0.7$, $P = 9E-25$; Lactotrope versus Thyrotrope: $\logFC = 0.4$, $P = 1.1E-08$). We have added P value bars to the violin plots of *ZBTB20* as well as other transcription factors in NEW Fig. 6c. We have also clarified the description in the new manuscript (L302).

9. *Fig. 7 presents the stepwise differentiation of gonadotropes. This is consistent with reports in mice. However, the diagram in Fig. 7e is confusing because it highlights transcription factors that may or may not be enriched for the stages shown.*

Response:

We have revised the diagram and only highlighted transcription factors that are enriched for the stages in NEW Fig. 7e (see also Figure R6).

Figure R6. Diagram of the development of human fetal anterior pituitary. TFs that are enriched for the stages are highlighted.

10. Fig. 8. The mechanism for comparing adult rodent gene expression data sets with the fetal human dataset seems problematic. It is also not clear why the number of human specific genes should be greater than rodent. Please discuss false negatives. What fraction of the reads did not map back to the genome?

Response:

We have re-analyzed the part of identifying the species-specific genes (relating to Fig. 8c). Fig. 8a and 8b nicely exhibited the expected conservation between the human and rodent pituitaries, and we kept the original analysis. We recognized cell type specific genes for each species and then identified human (or fetal)-specific genes and rodent (or adult)-specific genes for the corticotropes, somatotropes, lactotropes, thyrotropes and gonadotropes, respectively; the rodent genes required to be both the mouse and the rat genes. Among the genes, *CGB* and *GH2* are primate-specific genes and thus do not exist in the rodent data (NEW Fig. 8c and Figure R7). We understand that, since the human dataset was fetal and the rodent datasets were adult, any differences between these two

datasets could reflect the species or stage differences. Indeed, we found that *Trhr* and *Esr1* were highly expressed in the rodent thyrotropes and lactotropes, respectively, but lowly expressed in the human fetal pituitary, thus reflecting stage differences between these two datasets (NEW Fig. 8c and Figure R7). Interestingly, we found that *AR* was highly expressed in the corticotropes of the human fetal pituitary while lowly expressed in the corticotropes and instead highly expressed in the gonatropes of the rodent adult pituitaries. Also, *PRLHR*, which was mainly expressed in the brain, was highly expressed in gonatropes of the human fetal pituitary while nearly not expressed in the rodent datasets (NEW Fig. 8c and Figure R7). *IL17RB* and *GAD2* also showed specific expression in the human fetal pituitary.

In the reanalysis, we have applied the RESCUE method to impute the dropouts for the rodent data. To reduce false results, we have required the rodent genes to be both the mouse and the rat genes in the new analysis, and have also required high expression levels of the human-specific genes. The differences between the numbers of human- and rodent-specific genes alleviated (NEW supplementary Table 7).

We used a plate-based STRT-seq method while the rodent data used a droplet-based 10X genomics scRNA-seq method. The 10X genomic scRNA-seq method has very high throughput. However, since the reverse transcription reaction is relatively low efficient on the bead, it recovers lower gene numbers comparing with our plate-based method which performs the reverse transcription reaction in solution. The mapping ratio of 10X genomics technique is higher than our method (~90% mapping ratio for the mouse data and ~50% for our data). However, our method covered more genes per cell comparing with the rodent data (4,506/2,140/2,705 genes per cell for human, mouse and rat datasets, NEW supplementary Fig. 7a). The higher efficiency of the plate-based method comparing with the droplet-based method has also been recently reported by Ding et al in Nature Biotechnology (PMID: 32341560).

We have added the description of the results in L446-L462, and the description of the method in L733-L737.

Figure R7. Violin plots of the representative distinct cell type-specific genes between the human fetal pituitary and rodent adult pituitaries.

11. *Supplemental Fig. 2 nicely illustrates the onset of expression of pituitary hormones during gestation (2b). Panel 2c shows the enrichment of the top 10 transcription factors for each cell type. Other than the stem cells and corticotropes, the enrichment for other cell types seems very noisy. This is also true for the comparison of the PIT1 lineages in Supplementary Figure 5. These figures contrast with Supplementary Fig. 6 where the gene expression profiling of the stages of gonadotropes seems more clear.*

Response:

The distinct expression pattern between Supplementary Fig. 2c, and Supplementary Fig. 5 was due to different algorithms for these figures. Top enriched TFs of Supplementary Fig. 2c were identified by SCENIC, a method combines the cis-regulatory motif and co-expression module analyses for predicting activated transcription factors and their putative regulatory subnetworks. The method scores the activity of the regulons (the regulatory subnetwork of the transcription factor), but not the expression level of the transcription factor. We considered that this information is unique for understanding the gene regulatory network. Since the genes in Supplementary Fig. 2c were ranked by the activity of the regulons of the TFs, but not their expression levels, it seemed noisy but actually provided valuable information beyond the expression level. The gene list of Supplementary Fig. 2c (displaying gene expression) was actually corresponding to Fig.

1e (displaying TF activity). We apologize that the description in the OLD manuscript was not clear and we have added the description in the legend of Fig. 1e and Supplementary Fig. 2c in the NEW manuscript.

Supplementary Fig. 5 was based on differentially expressed genes between any two lineages of the somatotrope, the lactotrope and the thyrotrope, but not the typical approach as for Supplementary Fig. 6 which compared between one cell type and all other cells. We have rearranged the figure to directly show the DEGs between any two lineages to make it more clear (NEW Supplementary Fig. 5b and Figure R8).

Figure R8. Heatmap of average expression of significantly differentially expressed TFs between each two cell types of PIT1 lineage.

12. Technical considerations:

• Is 4000 cells enough? Does it make sense that there are 4506 genes expressed and ~86K transcripts?

Response:

Whether the number of cells is enough or not depends on the proportion of the target cell subpopulation within the whole population. If a target cell subpopulation is expected to be only 0.1%, 4,000 cells are not enough to recover this cell subpopulation, since you are expected to get only 4 cells which are not able to form a subcluster. In this study, the main cell subpopulations, including the stem cells and the five terminal endocrine cell types, typically reached several hundred cells. The progenitor and precursor cells, including the Pro.PIT_all, Pre.Thy and Pre.Gonado, reached about one hundred cells. Thus, 4,000 cells were enough for characterizing these cell populations. However, 4,000 cells were not enough for identifying some rare cell types such as the melanocyte. We have added related descriptions in L269-L275.

The number of 4,506 genes per cell is good as for a scRNA-seq technique. We used a plate-based STRT-seq method. The method covered more genes per cell comparing with the recently published mouse and rodent adult pituitary scRNA-seq data using the droplet-based 10X genomics method (4,506/ 2,140/ 2,705 genes per cell for human, mouse and rat datasets, NEW supplementary Fig. 7a). The higher efficiency of the plate-based method comparing with the droplet-based method has also been recently reported by Ding et al in Nature Biotechnology (PMID: 32341560).

13 Does the correction for batch effects make sense? In the supplement they show similar GAPDH expression across sets.

Response:

Batch effects are complex and Supplementary Fig. 1b only addressed the batch effects on certain aspects and thus we cannot claim that these results can demonstrate minimal batch effects in all aspects. Thus we have more directly described the results as “The samples

gave similar gene numbers and similar expression levels of *GAPDH* across batches” in L86, and “Each cell cluster was composed of multiple fetal samples, and the samples of similar stages, or different sexes, were largely mixedly distributed” in L109-L110.

14. How was the stem cell reclustering done? Was expression of SOX2 the basis for calling these stem cells?

Response:

The stem cell cluster were recognized by several marker genes including *SOX2*, *PROP1*, *SOX9*, *LHX3* and *HES1* (Fig. 1e and Fig2a). The stem cells were reclustered by Seurat by setting the resolution parameter as 0.3. We have added the method details in the L663.

Minor things:

15. L. 166 refers to *WNT5A* as canonical WNT signaling, but it usually acts in a non-canonical fashion.

Response:

Thanks. We have corrected it in the new manuscript (L176).

16. Line 187 cadeharin – cadherin

Response:

Thanks. We have corrected it in the new manuscript (L203).

17. l. 248 satges – stages

Response:

Thanks. We have corrected it in the new manuscript (L252).

18. l. 325 *neruoD4 – neuroD4*

Response:

Thanks. We have corrected it in the new manuscript (L330).

19. *Present Zbtb20 expression as ubiquitous yet lactotrope specific. Please clarify.*

Response:

ZBTB20 was expressed in all pituitary cell types. The expression level was slightly but significantly higher in the lactotrope comparing with the somatotrope and the thyrotrope (Lactotrope versus Somatotrope: logFC = 0.7, $P = 9E-25$; Lactotrope versus Thyrotrope: logFC = 0.4, $P = 1.1E-08$). We have clarified this in L307.

20. l. 309 *Gata2 ko references Dasen et al., which described the effects of a dominant negative Gata2 transgene and ectopic Gata2 expression. This is a valid reference. The authors ref. 49 is the Gata2 ko.*

Response:

We have described the *Gata2* studies in detail in the Discussion section, L488. We have simplified the sentence here in L343. We also have corrected the reference in the new manuscript (L311).

21. *In the discussion of mammo-somatotropes Nasonkin et al., is cited as a reference. This paper does not present evidence for cells expressing both prolactin and growth hormone.*

Response:

We have deleted this reference in the new manuscript (L369).

22. *There are numerous errors in the listing of authors in the references.*

Response:

We have checked and corrected them.

23. Supplemental Fig. 1d. Is there a better way to illustrate males and females? The circles and triangles are difficult to discern.

Response:

We have added NEW Supplementary Fig. 1e and Supplementary Fig. 2a to clearly illustrate males and females (see also Figure R9).

Figure R9. Distribution of the female and male samples on all pituitary cells (left) and endocrine cells (right) shown in the UMAP plot.

24. Supplemental Fig. 1 g. presents 8 disease terms for 19 genes. It would be more accurate to describe these.

Response:

We have revised to describe the pituitary phenotypes including Combined pituitary hormone deficiency (CPHD), isolated GH deficiency (IGHD) and as hypogonadotropic

hypogonadism (HH) in the NEW Supplementary Fig. 1g (see also Figure R10).

Figure R10 Heatmap of the averaged z-scored expression (red, high; blue, low) of TFs related pituitary diseases in each cell type. The pituitary phenotypes are shown including combined pituitary hormone deficiency (CPHD), isolated GH deficiency (IGHD) and hypogonadotropic hypogonadism (HH).

Reviewers' Comments:

Reviewer #1:

Remarks to the Author:

The new trajectories look to be improved in several ways now appear more consistent with RNA velocity. The figure 6f heatmap appears identical to the prior version of manuscript - should it be updated?

While it is true that the new trajectories appear improved, the corticotroph trajectory still crosses pro-PIT lineage, and near to pre-gonadotroph. The intention of this comment was to point out that at the earliest sample, 7Wks, corticotroph cells were already a separate cluster, distinct from Stem cells and cell cycle cells (Yellow points in Fig. 1d). This suggests the origin point of corticotroph trajectory should not be Stem 1. The portion of the trajectory between Stem 1 and corticotroph cluster is probably not reliable, including the appearance of a small number of gonadotrophs along the corticotroph lineage in Fig. 4c. It may be difficult to address this in the Slingshot analysis, but this fact should at least be acknowledged in the manuscript.

For all other cell types, the new trajectories beginning at Stem 1 appear reasonable.

Reviewer #2:

Remarks to the Author:

The authors have taken the reviews to heart and responded very appropriately by providing additional data analysis, revised figures and adjusted text as needed. There are some minor grammatical errors which can be dealt with at the editing stage. I have no concerns with the current manuscript.

REVIEWERS' COMMENTS:

Reviewer #1 (Remarks to the Author):

The new trajectories look to be improved in several ways now appear more consistent with RNA velocity. The figure 6f heatmap appears identical to the prior version of manuscript - should it be updated?

Response: The authors thank the reviewer for the positive comments and suggestions. For

Fig. 6f, we have updated with the new trajectory analysis in the new version of main figure.

Figure 6f Heatmap showing the relative expression levels of the significantly upregulated TFs

of thyrotropes along the pseudotime axis. Colors: loess-smoothed expression (red, high; blue, low). The columns represent cells being ordered along the pseudotime axis, and cell type information is shown above the heatmap. Rows represent genes being ordered by their peak expression along the pseudotime axis.

While it is true that the new trajectories appear improved, the corticotroph trajectory still crosses pro-PIT lineage, and near to pre-gonadotroph. The intention of this comment was to point out that at the earliest sample, 7Wks, corticotroph cells were already a separate cluster, distinct from Stem cells and cell cycle cells (Yellow points in Fig. 1d). This suggests the origin point of corticotroph trajectory should not be Stem 1. The portion of the trajectory between Stem 1 and corticotroph cluster is probably not reliable, including the appearance of a small number of gonadotrophs along the corticotroph lineage in Fig. 4c. It may be difficult to address this in the Slingshot analysis, but this fact should at least be acknowledged in the manuscript.

For all other cell types, the new trajectories beginning at Stem 1 appear reasonable.

Response: We thank the reviewer for the suggestions and comments. We agree that corticotrope cells in our data may be already a separate cluster, which may differentiate from stem cells earlier than Stem1 we collected in the earliest sample (7 weeks). We acknowledged this in the new manuscript, L719-721.

Reviewer #2 (Remarks to the Author):

The authors have taken the reviews to heart and responded very appropriately by providing additional data analysis, revised figures and adjusted text as needed. There are some minor grammatical errors which can be dealt with at the editing stage. I have no concerns with the current manuscript.

Response: The authors thank the reviewer for the positive comments.